# A Unified Framework for Uniform Signal Recovery in Nonlinear Generative Compressed Sensing

**Junren Chen**[*]
University of Hong Kong
chenjr58@connect.hku.hk

**Jonathan Scarlett**
National University of Singapore
scarlett@comp.nus.edu.sg

**Michael K. Ng**
Hong Kong Baptist University
michael-ng@hkbu.edu.hk

**Zhaoqiang Liu**[*]
UESTC
zqliu12@gmail.com

## Abstract

In generative compressed sensing (GCS), we want to recover a signal $\boldsymbol{x}^* \in \mathbb{R}^n$ from $m$ measurements ($m \ll n$) using a generative prior $\boldsymbol{x}^* \in G(\mathbb{B}_2^k(r))$, where $G$ is typically an $L$-Lipschitz continuous generative model and $\mathbb{B}_2^k(r)$ represents the radius-$r$ $\ell_2$-ball in $\mathbb{R}^k$. Under nonlinear measurements, most prior results are non-uniform, i.e., they hold with high probability for a fixed $\boldsymbol{x}^*$ rather than for all $\boldsymbol{x}^*$ simultaneously. In this paper, we build a unified framework to derive uniform recovery guarantees for nonlinear GCS where the observation model is nonlinear and possibly discontinuous or unknown. Our framework accommodates GCS with 1-bit/uniformly quantized observations and single index models as canonical examples. Specifically, using a single realization of the sensing ensemble and generalized Lasso, *all* $\boldsymbol{x}^* \in G(\mathbb{B}_2^k(r))$ can be recovered up to an $\ell_2$-error at most $\epsilon$ using roughly $\tilde{O}(k/\epsilon^2)$ samples, with omitted logarithmic factors typically being dominated by $\log L$. Notably, this almost coincides with existing non-uniform guarantees up to logarithmic factors, hence the uniformity costs very little. As part of our technical contributions, we introduce the Lipschitz approximation to handle discontinuous observation models. We also develop a concentration inequality that produces tighter bounds for product processes whose index sets have low metric entropy. Experimental results are presented to corroborate our theory.

## 1 Introduction

In compressed sensing (CS) that concerns the reconstruction of low-complexity signals (typically sparse signals) [5, 6, 15], it is standard to employ a random measurement ensemble, i.e., a random sensing matrix and other randomness that produces the observations. Thus, a recovery guarantee involving a single draw of the measurement ensemble could be *non-uniform* or *uniform* — the non-uniform one ensures the accurate recovery of any fixed signal with high probability, while the uniform one states that one realization of the measurements works simultaneously for all structured signals of interest. Uniformity is a highly desired property in CS, since in applications the measurement ensemble is typically fixed and should work for all signals [17]. Besides, the derivation of a uniform guarantee is often significantly harder than a non-uniform one, making uniformity an interesting theoretical problem in its own right.

---

[*]Corresponding authors.

37th Conference on Neural Information Processing Systems (NeurIPS 2023).

Inspired by the tremendous success of deep generative models in different applications, it was recently proposed to use a generative prior to replace the commonly used sparse prior in CS [2], which led to numerical success such as a significant reduction of the measurement number. This new perspective for CS, which we call generative compressed sensing (GCS), has attracted a large volume of research interest, e.g., nonlinear GCS [29, 33, 45], MRI applications [24, 46], and information-theoretic bounds [27, 34], among others. This paper focuses on the uniform recovery problem for nonlinear GCS, which is formally stated below. Our main goal is to build a unified framework that can produce uniform recovery guarantees for various nonlinear measurement models.

**Problem:** Let $\mathbb{B}_2^k(r)$ be the $\ell_2$-ball with radius $r$ in $\mathbb{R}^k$. Suppose that $G : \mathbb{B}_2^k(r) \to \mathbb{R}^n$ is an $L$-Lipschitz continuous generative model, $\boldsymbol{a}_1, ..., \boldsymbol{a}_m \in \mathbb{R}^n$ are the sensing vectors, $\boldsymbol{x}^* \in \mathcal{K} := G(\mathbb{B}_2^k(r))$ is the underlying signal, and we have the observations $y_i = f_i(\boldsymbol{a}_i^\top \boldsymbol{x}^*)$, $i = 1, \ldots, m$, where $f_1(\cdot), \ldots, f_m(\cdot)$ are possibly unknown,[2] possibly random non-linearities. Given a single realization of $\{\boldsymbol{a}_i, f_i\}_{i=1}^m$, under what conditions can we *uniformly* recover all $\boldsymbol{x}^* \in \mathcal{K}$ from the corresponding $\{\boldsymbol{a}_i, y_i\}_{i=1}^m$ up to an $\ell_2$-norm error of $\epsilon$?

## 1.1 Related Work

We divide the related works into nonlinear CS (based on traditional structures like sparsity) and nonlinear GCS.

**Nonlinear CS:** Beyond the standard linear CS model where one observes $y_i = \boldsymbol{a}_i^\top \boldsymbol{x}^*$, recent years have witnessed rapidly increasing literature on nonlinear CS. An important nonlinear CS model is 1-bit CS that only retains the sign $y_i = \text{sign}(\boldsymbol{a}_i^\top \boldsymbol{x}^*)$ [3, 22, 41, 42]. Subsequent works also considered 1-bit CS with dithering $y_i = \text{sign}(\boldsymbol{a}_i^\top \boldsymbol{x}^* + \tau_i)$ to achieve norm reconstruction under sub-Gaussian sensing vectors [9, 14, 48]. Besides, the benefit of using dithering was found in uniformly quantized CS with observation $y_i = \mathcal{Q}_\delta(\boldsymbol{a}_i^\top \boldsymbol{x}^* + \tau_i)$, where $\mathcal{Q}_\delta(\cdot) = \delta(\lfloor \frac{\cdot}{\delta} \rfloor + \frac{1}{2})$ is the uniform quantizer with resolution $\delta$ [8, 48, 52]. Moreover, the authors of [16, 43, 44] studied the more general single index model (SIM) where the observation $y_i = f_i(\boldsymbol{a}_i^\top \boldsymbol{x}^*)$ involves (possibly) unknown nonlinearity $f_i$.

While the restricted isometry property (RIP) of the sensing matrix $\boldsymbol{A} = [\boldsymbol{a}_1, ..., \boldsymbol{a}_m]^\top$ leads to uniform recovery in linear CS [4, 15, 49], this is not true in nonlinear CS. In fact, many existing results are non-uniform [9, 16, 21, 41, 43, 44, 48], and some uniform guarantees can be found in [7, 8, 14, 17, 41, 42, 52]. Most of these uniform guarantees suffer from a slower error rate.

The most relevant work to this paper is the recent work [17] that described a unified approach to uniform signal recovery for nonlinear CS. The authors of [17] showed that in the aforementioned models with $k$-sparse $\boldsymbol{x}^*$, a uniform $\ell_2$-norm recovery error of $\epsilon$ could be achieved via generalized Lasso using roughly $k/\epsilon^4$ measurements [17, Section 4]. In this work, we build a unified framework for uniform signal recovery in nonlinear GCS. To achieve a uniform $\ell_2$-norm error of $\epsilon$ in the above models with the generative prior $\boldsymbol{x}^* \in G(\mathbb{B}_2^k(r))$, our framework only requires a number of samples proportional to $k/\epsilon^2$. Unlike [17] that used the technical results [36] to bound the product process, we develop a concentration inequality that produces a tighter bound in the setting of generative prior, thus allowing us to derive a sharper uniform error rate.

**Nonlinear GCS:** Building on the seminal work by Bora *et al.* [2], numerous works have investigated linear or nonlinear GCS [1, 11, 12, 19, 20, 23, 25, 30, 39, 40, 51], with a recent survey [47] providing a comprehensive overview. Particularly for nonlinear GCS, 1-bit CS with generative models has been studied in [26, 31, 45], and generative priors have been used for SIM in [29, 32, 33]. In addition, score-based generative models have been applied to nonlinear CS in [10, 38].

The majority of research for nonlinear GCS focuses on non-uniform recovery, with only a few exceptions [33, 45]. Specifically, under a generative prior, [33, Section 5] presented uniform recovery guarantees for SIM where $y_i = f_i(\boldsymbol{a}_i^\top \boldsymbol{x}^*)$ with deterministic Lipschitz $f_i$ or $f_i(x) = \text{sign}(x)$. Their proof technique is based on the local embedding property developed in [31], which is a geometric property that is often problem-dependent and currently only known for 1-bit measurements and deterministic Lipschitz link functions. In contrast, our proof technique does not rely on such

---

[2]In order to establish a unified framework, our recovery method (2.1) involves a parameter $T$ that should be chosen according to $f_i$. For the specific single index model with possibly unknown $f_i$, we can follow prior works [33, 43] to assume that $T\boldsymbol{x}^* \in \mathcal{K}$, and recover $\boldsymbol{x}^*$ without using $T$. See Remark 5 for more details.

geometric properties and yields a unified framework with more generality. Furthermore, [33] did not consider dithering, which limits their ability to estimate the norm of the signal.

The authors of [45] derived a uniform guarantee from dithered 1-bit measurements under bias-free ReLU neural network generative models, while we obtain a uniform guarantee with the comparable rate for more general Lipschitz generative models. Additionally, their recovery program differs from the generalized Lasso approach (*cf.* Section 2.1) used in our work. Specifically, they minimize an $\ell_2$ loss with $\|x\|_2^2$ as the quadratic term, while generalized Lasso uses $\|Ax\|_2^2$ that depends on the sensing vector. As a result, our approach can be readily generalized to sensing vectors with an unknown covariance matrix [33, Section 4.2], unlike [45] that is restricted to isotropic sensing vectors. Under random dithering, while [45] only considered 1-bit measurements, we also present new results for uniformly quantized measurements (also referred to as multi-bit quantizer in some works [13]).

## 1.2 Contributions

In this paper, we build a unified framework for uniform signal recovery in nonlinear GCS. We summarize the paper structure and our main contributions as follows:

- We present Theorem 1 as our main result in Section 2. Under rather general observation models that can be discontinuous or unknown, Theorem 1 states that the uniform recovery of all $x^* \in G(\mathbb{B}_2^k(r))$ up to an $\ell_2$-norm error of $\epsilon$ can be achieved using roughly $O\left(\frac{k \log L}{\epsilon^2}\right)$ samples. Specifically, we obtain uniform recovery guarantees for 1-bit GCS, 1-bit GCS with dithering, Lipschitz-continuous SIM, and uniformly quantized GCS with dithering.
- We provide a proof sketch in Section 3. Without using the embedding property as in [33], we handle the discontinuous observation model by constructing a Lipschitz approximation. Compared to [17], we develop a new concentration inequality (Theorem 2) to derive tighter bounds for the product processes arising in the proof.

We also perform proof-of-concept experiments on the MNIST [28] and CelebA [35] datasets for various nonlinear models to demonstrate that by using a single realization of $\{a_i, f_i\}_{i=1}^m$, we can obtain reasonably accurate reconstruction for multiple signals. Due to the page limit, the experimental results and detailed proofs are provided in the supplementary material.

## 1.3 Notation

We use boldface letters to denote vectors and matrices, while regular letters are used for scalars. For a vector $x$, we let $\|x\|_q$ ($1 \le q \le \infty$) denote its $\ell_q$-norm. We use $\mathbb{B}_q^n(r) := \{z \in \mathbb{R}^n : \|z\|_q \le r\}$ to denote the $\ell_q$ ball in $\mathbb{R}^n$, and $(\mathbb{B}_q^n(r))^c$ represents its complement. The unit Euclidean sphere is denoted by $\mathbb{S}^{n-1} := \{x \in \mathbb{R}^n : \|x\|_2 = 1\}$. We use $C, C_i, c_i, c$ to denote absolute constants whose values may differ from line to line. We write $A = O(B)$ or $A \lesssim B$ (resp. $A = \Omega(B)$ or $A \gtrsim B$) if $A \le CB$ for some $C$ (resp. $A \ge cB$ for some $c$). We write $A \asymp B$ if $A = O(B)$ and $A = \Omega(B)$ simultaneously hold. We sometimes use $\tilde{O}(\cdot)$ to further hide logarithmic factors, where the hidden factors are typically dominated by $\log L$ in GCS, or $\log n$ in CS. We let $\mathcal{N}(\mu, \Sigma)$ be the Gaussian distribution with mean $\mu$ and covariance matrix $\Sigma$. Given $\mathcal{K}_1, \mathcal{K}_2 \subset \mathbb{R}^n$, $a \in \mathbb{R}^n$ and some $a \in \mathbb{R}$, we define $\mathcal{K}_1 \pm \mathcal{K}_2 := \{x_1 \pm x_2 : x_1 \in \mathcal{K}_1, x_2 \in \mathcal{K}_2\}$, $a + \mathcal{K}_1 := \{a\} + \mathcal{K}_1$, and $a\mathcal{K}_1 := \{ax : x \in \mathcal{K}_1\}$. We also adopt the conventions of $a \wedge b = \min\{a, b\}$, and $a \vee b = \max\{a, b\}$.

## 2 Main Results

We first give some preliminaries.

**Definition 1.** *For a random variable $X$, we define the sub-Gaussian norm $\|X\|_{\psi_2} := \inf\{t > 0 : \mathbb{E}\exp(X^2/t^2) \le 2\}$ and the sub-exponential norm $\|X\|_{\psi_1} := \inf\{t > 0 : \mathbb{E}\exp(|X|/t) \le 2\}$. $X$ is sub-Gaussian (resp. sub-exponential) if $\|X\|_{\psi_2} < \infty$ (resp. $\|X\|_{\psi_1} < \infty$). For a random vector $x \in \mathbb{R}^n$, we let $\|x\|_{\psi_2} := \sup_{v \in \mathbb{S}^{n-1}} \|v^\top x\|_{\psi_2}$.*

**Definition 2.** *Let $\mathcal{S}$ be a subset of $\mathbb{R}^n$. We say that a subset $\mathcal{S}_0 \subset \mathcal{S}$ is an $\eta$-net of $\mathcal{S}$ if every point in $\mathcal{S}$ is at most $\eta$ distance away from some point in $\mathcal{S}_0$, i.e., $\mathcal{S} \subset \mathcal{S}_0 + \mathbb{B}_2^n(\eta)$. Given a radius $\eta$, we*

*define the covering number $\mathscr{N}(\mathcal{S}, \eta)$ as the minimal cardinality of an $\eta$-net of $\mathcal{S}$. The metric entropy of $\mathcal{S}$ with respect to radius $\eta$ is defined as $\mathscr{H}(\mathcal{S}, \eta) = \log \mathscr{N}(\mathcal{S}, \eta)$.*

## 2.1 Problem Setup

We make the following assumptions on the observation model.

**Assumption 1.** *Let $\boldsymbol{a} \sim \mathcal{N}(0, \boldsymbol{I}_n)$ and let $f$ be a possibly unknown, possibly random non-linearity that is independent of $\boldsymbol{a}$. Let $(\boldsymbol{a}_i, f_i)_{i=1}^m$ be i.i.d. copies of $(\boldsymbol{a}, f)$. With a single draw of $(\boldsymbol{a}_i, f_i)_{i=1}^m$, for $\boldsymbol{x}^* \in \mathcal{K} = G(\mathbb{B}_2^k(r))$, where $G : \mathbb{B}_2^k(r) \to \mathbb{R}^n$ is an L-Lipschitz generative model, we observe $\left\{ y_i := f_i(\boldsymbol{a}_i^\top \boldsymbol{x}^*) \right\}_{i=1}^m$. We can express the model more compactly as $\boldsymbol{y} = \boldsymbol{f}(\boldsymbol{A}\boldsymbol{x}^*)$, where $\boldsymbol{A} = [\boldsymbol{a}_1, ..., \boldsymbol{a}_m]^\top \in \mathbb{R}^{m \times n}$, $\boldsymbol{f} = (f_1, ..., f_m)^\top$ and $\boldsymbol{y} = (y_1, ..., y_m)^\top \in \mathbb{R}^m$.*

In this work, we consider the generalized Lasso as the recovery method [16, 33, 43], whose core idea is to ignore the non-linearity and minimize the regular $\ell_2$ loss. In addition, we need to specify a constraint that reflects the low-complexity nature of $\boldsymbol{x}^*$, and specifically, we introduce a problem-dependent scaling factor $T \in \mathbb{R}$ and use the constraint "$\boldsymbol{x} \in T\mathcal{K}$". Note that this is necessary even if the problem is linear; for example, with observations $\boldsymbol{y} = 2\boldsymbol{A}\boldsymbol{x}^*$, one needs to minimize the $\ell_2$ loss over "$\boldsymbol{x} \in 2\mathcal{K}$". Also, when the generative prior is given by $T\boldsymbol{x}^* \in \mathcal{K} = G(\mathbb{B}_2^k(r))$, we should simply use "$\boldsymbol{x} \in \mathcal{K}$" as constraint; this is technically equivalent to the treatment adopted in [33] (see more discussions in Remark 5 below). Taken collectively, we consider

$$\hat{\boldsymbol{x}} = \arg \min_{\boldsymbol{x} \in T\mathcal{K}} \|\boldsymbol{y} - \boldsymbol{A}\boldsymbol{x}\|_2. \tag{2.1}$$

Importantly, we want to achieve uniform recovery of all $\boldsymbol{x}^* \in \mathcal{K}$ with a single realization of $(\boldsymbol{A}, \boldsymbol{f})$.

## 2.2 Assumptions

Let $f$ be the function that characterizes our nonlinear measurements. We introduce several assumptions on $f$ here, and then verify them for specific models in Section 2.3. We define the set of discontinuities as

$$\mathscr{D}_f = \{a \in \mathbb{R} : f \text{ is discontinuous at } a\}.$$

We define the notion of jump discontinuity as follows.

**Definition 3.** (Jump discontinuity). *A function $f : \mathbb{R} \to \mathbb{R}$ has a jump discontinuity at $x_0$ if both $L^- := \lim_{x \to x_0^-} f(x)$ and $L^+ := \lim_{x \to x_0^+} f(x)$ exist but $L^- \neq L^+$. We simply call the oscillation at $x_0$, i.e., $|L^+ - L^-|$, the jump.*

Roughly put, our framework applies to piece-wise Lipschitz continuous $f_i$ with (at most) countably infinite jump discontinuities, which have bounded jumps and are well separated. The precise statement is given below.

**Assumption 2.** *For some $(B_0, L_0, \beta_0)$, the following statement unconditionally holds true for any realization of $f$ (specifically, $f_1, \ldots, f_m$ in our observations):*

- *$\mathscr{D}_f$ is one of the following: $\varnothing$, a finite set, or a countably infinite set;*
- *All discontinuities of $f$ (if any) are jump discontinuities with the jump bounded by $B_0$;*
- *$f$ is $L_0$-Lipschitz on any interval $(a, b)$ satisfying $(a, b) \cap \mathscr{D}_f = \varnothing$.*
- *$|a - b| \geq \beta_0$ holds for any $a, b \in \mathscr{D}_f$, $a \neq b$ (we set $\beta_0 = \infty$ if $|\mathscr{D}_f| \leq 1$).*

*For simplicity, we assume $f(x_0) = \lim_{x \to x_0^+} f(x)$ for $x_0 \in \mathscr{D}_f$.[3]*

We note that Assumption 2 is satisfied by $L$-Lipschitz $f$ with $(B_0, L_0, \beta_0) = (0, L, \infty)$, 1-bit quantized observation $f(\cdot) = \text{sign}(\cdot + \tau)$ ($\tau$ is the potential dither, similarly below) with $(B_0, L_0, \beta_0) = (2, 0, \infty)$, and uniformly quantized observation $f(\cdot) = \delta\left(\lfloor \frac{\cdot + \tau}{\delta} \rfloor + \frac{1}{2}\right)$ with $(B_0, L_0, \beta_0) = (\delta, 0, \delta)$.

---

[3]This is very mild because the observations are $f_i(\boldsymbol{a}_i^\top \boldsymbol{x})$, while $\mathbb{P}(\boldsymbol{a}^\top \boldsymbol{x} \in \mathscr{D}_{f_i}) = 0$ (as $\mathscr{D}_{f_i}$ is at most countably infinite and $\boldsymbol{a} \sim \mathcal{N}(0, \boldsymbol{I}_n)$).

Under Asssumption 2, for any $\beta \in [0, \frac{\beta_0}{2})$ we construct $f_{i,\beta}$ as the Lipschitz approximation of $f_i$ to deal with the potential discontinuity of $f_i$ (i.e., $\mathscr{D}_{f_i} \neq \varnothing$). Specifically, $f_{i,\beta}$ modifies $f_i$ in $\mathscr{D}_{f_i} + [-\frac{\beta}{2}, \frac{\beta}{2}]$ to be piece-wise linear and Lipschitz continuous; see its precise definition in (3.4).

We develop Theorem 2 to bound certain product processes appearing in the analysis, which produces bounds tighter than [36] when the index sets have low metric entropy. To make Theorem 2 applicable, we further make the following Assumption 3, which can be checked case-by-case by estimating the sub-Gaussian norm and probability tail. Also, $U_g^{(1)}$ and $U_g^{(2)}$ can even be a bit crude because the measurement number in Theorem 1 depends on them in a logarithmic manner.

**Assumption 3.** *Let $\boldsymbol{a} \sim \mathcal{N}(0, \boldsymbol{I}_n)$, under Assumptions 1-2, we define the Lipschitz approximation $f_{i,\beta}$ as in (3.4). We let*

$$\xi_{i,\beta}(a) := f_{i,\beta}(a) - Ta, \ \varepsilon_{i,\beta}(a) := f_{i,\beta}(a) - f_i(a). \tag{2.2}$$

*For all $\beta \in (0, \frac{\beta_0}{2})$, we assume the following holds with some parameters $(A_g^{(1)}, U_g^{(1)}, P_0^{(1)})$ and $(A_g^{(2)}, U_g^{(2)}, P_0^{(2)})$:*

- $\sup_{\boldsymbol{x} \in \mathcal{K}} \|\xi_{i,\beta}(\boldsymbol{a}^\top \boldsymbol{x})\|_{\psi_2} \leq A_g^{(1)}$, $\mathbb{P}\big( \sup_{\boldsymbol{x} \in \mathcal{K}} |\xi_{i,\beta}(\boldsymbol{a}^\top \boldsymbol{x})| \leq U_g^{(1)} \big) \geq 1 - P_0^{(1)}$;
- $\sup_{\boldsymbol{x} \in \mathcal{K}} \|\varepsilon_{i,\beta}(\boldsymbol{a}^\top \boldsymbol{x})\|_{\psi_2} \leq A_g^{(2)}$, $\mathbb{P}\big( \sup_{\boldsymbol{x} \in \mathcal{K}} |\varepsilon_{i,\beta}(\boldsymbol{a}^\top \boldsymbol{x})| \leq U_g^{(2)} \big) \geq 1 - P_0^{(2)}$.

To build a more complete theory we further introduce two useful quantities. For some $\boldsymbol{x} \in \mathcal{K}$, we define the target mismatch $\rho(\boldsymbol{x})$ as in [17, Definition 1]:

$$\rho(\boldsymbol{x}) = \big\| \mathbb{E}\big[ f_i(\boldsymbol{a}_i^\top \boldsymbol{x}) \boldsymbol{a}_i \big] - T\boldsymbol{x} \big\|_2. \tag{2.3}$$

It is easy to see that $\mathbb{E}\big[ f_i(\boldsymbol{a}_i^\top \boldsymbol{x}) \boldsymbol{a}_i \big]$ minimizes the expected $\ell_2$ loss $\mathbb{E}\big[ \|\boldsymbol{y} - \boldsymbol{A}\boldsymbol{x}\|_2^2 \big]$, thus one can roughly understand $\mathbb{E}\big[ f_i(\boldsymbol{a}_i^\top \boldsymbol{x}) \boldsymbol{a}_i \big]$ as the expectation of $\hat{\boldsymbol{x}}$. Since $T\boldsymbol{x}$ is the desired ground truth, a small $\rho(\boldsymbol{x})$ is intuitively an important ingredient for generalized Lasso to succeed. Fortunately, in many models, $\rho(\boldsymbol{x})$ with a suitably chosen $T$ will vanish (e.g., linear model [2], single index model [33], 1-bit model [31]) or at least be sufficiently small (e.g., 1-bit model with dithering [45]).

As mentioned before, our method to deal with discontinuity of $f_i$ is to introduce its approximation $f_{i,\beta}$, which differs from $f_i$ only in $\mathscr{D}_{f_i} + [-\frac{\beta}{2}, \frac{\beta}{2}]$. This will produce some bias because the actual observation is $f_i(\boldsymbol{a}_i^\top \boldsymbol{x}^*)$ rather than $f_{i,\beta}(\boldsymbol{a}_i^\top \boldsymbol{x}^*)$. Hence, for some $\boldsymbol{x} \in \mathcal{K}$ we define the following quantity to measure the bias induced by $f_{i,\beta}$:

$$\mu_\beta(\boldsymbol{x}) = \mathbb{P}\Big( \boldsymbol{a}^\top \boldsymbol{x} \in \mathscr{D}_{f_i} + \Big[ -\frac{\beta}{2}, \frac{\beta}{2} \Big] \Big), \ \boldsymbol{a} \sim \mathcal{N}(0, \boldsymbol{I}_n). \tag{2.4}$$

The following assumption can often be satisfied by choosing suitable $T$ and sufficiently small $\beta_1$.

**Assumption 4.** *Suppose Assumptions 1-3 hold true with parameters $B_0, L_0, \beta_0, A_g^{(1)}, A_g^{(2)}$. For the $T$ used in (2.1), $\rho(\boldsymbol{x})$ defined in (2.3) satisfies*

$$\sup_{\boldsymbol{x} \in \mathcal{K}} \rho(\boldsymbol{x}) \lesssim (A_g^{(1)} \vee A_g^{(2)}) \sqrt{\frac{k}{m}}. \tag{2.5}$$

*Moreover, there exists some $0 < \beta_1 < \frac{\beta_0}{2}$ such that*

$$(L_0 \beta_1 + B_0) \sup_{\boldsymbol{x} \in \mathcal{K}} \sqrt{\mu_{\beta_1}(\boldsymbol{x})} \lesssim (A_g^{(1)} \vee A_g^{(2)}) \sqrt{\frac{k}{m}}. \tag{2.6}$$

In the proof, the estimation error $\|\hat{\boldsymbol{x}} - T\boldsymbol{x}^*\|$ is contributed by a concentration term of scaling $\tilde{O}\big( (A_g^{(1)} \vee A_g^{(2)}) \sqrt{k/m} \big)$ and some bias terms. The main aim of Assumption 4 is to pull down the bias terms so that the concentration term is dominant.

## 2.3 Main Theorem and its Implications

We now present our general theorem and apply it to some specific models.

**Theorem 1.** *Under Assumptions 1-4, given any recovery accuracy $\epsilon \in (0,1)$, if it holds that $m \gtrsim (A_g^{(1)} \vee A_g^{(2)})^2 \frac{k\mathscr{L}}{\epsilon^2}$, then with probability at least $1 - m(P_0^{(1)} + P_0^{(2)}) - m\exp(-\Omega(n)) - C\exp(-\Omega(k))$ on a single realization of $(\boldsymbol{A}, \boldsymbol{f}) := (\boldsymbol{a}_i, f_i)_{i=1}^m$, we have the uniform signal recovery guarantee $\|\hat{\boldsymbol{x}} - T\boldsymbol{x}^*\|_2 \le \epsilon$ for all $\boldsymbol{x}^* \in \mathcal{K}$, where $\hat{\boldsymbol{x}}$ is the solution to (2.1) with $\boldsymbol{y} = \boldsymbol{f}(\boldsymbol{A}\boldsymbol{x}^*)$, and $\mathscr{L} = \log \widetilde{P}$ is a logarithmic factor with $\widetilde{P}$ being polynomial in $(L, n)$ and other parameters that typically scale as $O(L + n)$. See (C.11) for the precise expression of $\mathscr{L}$.*

To illustrate the power of Theorem 1, we specialize it to several models to obtain concrete uniform signal recovery results. Starting with Theorem 1, the remaining work is to select parameters that justify Assumptions 2-4. We summarize the strategy as follows: (i) Determine the parameters in Assumption 2 by the measurement model; (ii) Set $T$ that verifies (2.5) (see Lemmas 8-11 for the following models); (iii) Set the parameters in Assumption 3, for which bounding the norm of Gaussian vector is useful; (iv) Set $\beta_1$ to guarantee (2.6) based on some standard probability argument. We only provide suitable parameters for the following concrete models due to space limit, while leaving more details to Appendix E.

**(A) 1-bit GCS.** Assume that we have the 1-bit observations $y_i = \text{sign}(\boldsymbol{a}_i^\top \boldsymbol{x}^*)$; then $f_i(\cdot) = f(\cdot) = \text{sign}(\cdot)$ satisfies Assumption 2 with $(B_0, L_0, \beta_0) = (2, 0, \infty)$. In this model, it is hopeless to recover the norm of $\|\boldsymbol{x}^*\|_2$; as done in previous work, we assume $\boldsymbol{x}^* \in \mathcal{K} \subset \mathbb{S}^{n-1}$ [31, Remark 1]. We set $T = \sqrt{2/\pi}$ and take the parameters in Assumption 3 as $A_g^{(1)} \asymp 1, U_g^{(1)} \asymp \sqrt{n}, P_0^{(1)} \asymp \exp(-\Omega(n)), A_g^{(2)} \asymp 1, U_g^{(2)} \asymp 1, P_0^{(2)} = 0$. We take $\beta = \beta_1 \asymp \frac{k}{m}$ to guarantee (2.6). With these choices, Theorem 1 specializes to the following:

**Corollary 1.** *Consider Assumption 1 with $f_i(\cdot) = \text{sign}(\cdot)$ and $\mathcal{K} \subset \mathbb{S}^{n-1}$, let $\epsilon \in (0, 1)$ be any given recovery accuracy. If $m \gtrsim \frac{k}{\epsilon^2} \log\left(\frac{Lr\sqrt{mn}}{\epsilon \wedge (k/m)}\right)$,[4] then with probability at least $1 - 2m\exp(-cn) - m\exp(-\Omega(k))$ on a single draw of $(\boldsymbol{a}_i)_{i=1}^m$, we have the uniform signal recovery guarantee $\|\hat{\boldsymbol{x}} - \sqrt{\frac{2}{\pi}}\boldsymbol{x}^*\|_2 \le \epsilon$ for all $\boldsymbol{x}^* \in \mathcal{K}$, where $\hat{\boldsymbol{x}}$ is the solution to (2.1) with $\boldsymbol{y} = \text{sign}(\boldsymbol{A}\boldsymbol{x}^*)$ and $T = \sqrt{\frac{2}{\pi}}$.*

**Remark 1.** *A uniform recovery guarantee for generalized Lasso in 1-bit GCS was obtained in [33, Section 5]. Their proof relies on the local embedding property in [31]. Note that such geometric property is often problem-dependent and highly nontrivial. By contrast, our argument is free of geometric properties of this kind.*

**Remark 2.** *For traditional 1-bit CS, [17, Corollary 2] requires $m \gtrsim \tilde{O}(k/\epsilon^4)$ to achieve uniform $\ell_2$-accuracy of $\epsilon$ for all $k$-sparse signals, which is inferior to our $\tilde{O}(k/\epsilon^2)$. This is true for all remaining examples. To obtain such a sharper rate, the key technique is to use our Theorem 2 (rather than [36]) to obtain tighter bound for the product processes, as will be discussed in Remark 8.*

**(B) 1-bit GCS with dithering.** Assume that the $\boldsymbol{a}_i^\top \boldsymbol{x}^*$ is quantized to 1-bit with dither[5] $\tau_i \overset{iid}{\sim} \mathscr{U}[-\lambda, \lambda]$ for some $\lambda$ to be chosen, i.e., we observe $y_i = \text{sign}(\boldsymbol{a}_i^\top \boldsymbol{x}^* + \tau_i)$. Following [45] we assume $\mathcal{K} \subset \mathbb{B}_2^n(R)$ for some $R > 0$. Here, using dithering allows the recovery of signal norm $\|\boldsymbol{x}^*\|_2$, so we do not need to assume $\mathcal{K} \subset \mathbb{S}^{n-1}$ as in Corollary 1. We set $\lambda = CR\sqrt{\log m}$ with sufficiently large $C$, and $T = \lambda^{-1}$. In Assumption 3, we take $A_g^{(1)} \asymp 1, U_g^{(1)} \asymp \sqrt{n}, P_0^{(1)} \asymp \exp(-\Omega(n)), A_g^{(2)} \asymp 1, U_g^{(2)} \asymp 1$, and $P_0^{(2)} = 0$. Moreover, we take $\beta = \beta_1 = \frac{\lambda k}{m}$ to guarantee (2.6). Now we can invoke Theorem 1 to get the following.

**Corollary 2.** *Consider Assumption 1 with $f_i(\cdot) = \text{sign}(\cdot + \tau_i)$, $\tau_i \sim \mathscr{U}[-\lambda, \lambda]$ and $\mathcal{K} \subset \mathbb{B}_2^n(R)$, and $\lambda = CR\sqrt{\log m}$ with sufficiently large $C$. Let $\epsilon \in (0, 1)$ be any given recovery accuracy. If $m \gtrsim \frac{k}{\epsilon^2} \log\left(\frac{Lr\sqrt{mn}}{\lambda(\epsilon \wedge (k/m))}\right)$, then with probability at least $1 - 2m\exp(-cn) - m\exp(-\Omega(k))$ on a single draw of $(\boldsymbol{a}_i, \tau_i)_{i=1}^m$, we have the uniform signal recovery guarantee $\|\hat{\boldsymbol{x}} - \lambda^{-1}\boldsymbol{x}^*\|_2 \le \epsilon$ for all $\boldsymbol{x}^* \in \mathcal{K}$, where $\hat{\boldsymbol{x}}$ is the solution to (2.1) with $\boldsymbol{y} = \text{sign}(\boldsymbol{A}\boldsymbol{x}^* + \boldsymbol{\tau})$ (here, $\boldsymbol{\tau} = [\tau_1, ..., \tau_m]^\top$) and $T = \lambda^{-1}$.*

**Remark 3.** *To our knowledge, the only related prior result is in [45, Theorem 3.2]. However, their result is restricted to ReLU networks. By contrast, we deal with the more general Lipschitz generative models; by specializing our result to the ReLU network that is typically $(n^{\Theta(d)})$-Lipschitz [2] (d is*

---

[4]Here and in other similar statements, we implicitly assume a large enough implied constant.

[5]Throughout this work, the random dither is independent of the $\{\boldsymbol{a}_i\}_{i=1}^m$.

*the number of layers), our error rate coincides with theirs up to a logarithmic factor. Additionally, as already mentioned in the Introduction Section, our result can be generalized to a sensing vector with an unknown covariance matrix, unlike theirs which is restricted to isotropic sensing vectors. The advantage of their result is in allowing sub-exponential sensing vectors.*

**(C) Lipschitz-continuous SIM with generative prior.** Assume that any realization of $f$ is unconditionally $\hat{L}$-Lipschitz, which implies Assumption 2 with $(B_0, L_0, \beta_0) = (0, \hat{L}, \infty)$. We further assume $\mathbb{P}(f(0) \leq \hat{B}) \geq 1 - P_0'$ for some $(\hat{B}, P_0')$. Because the norm of $\boldsymbol{x}^*$ is absorbed into the unknown $f(\cdot)$, we assume $\mathcal{K} \subset \mathbb{S}^{n-1}$. We set $\beta = 0$ so that $f_{i,\beta} = f_i$. We introduce the quantities $\mu = \mathbb{E}[f(g)g], \psi = \|f(g)\|_{\psi_2}$, where $g \sim \mathcal{N}(0,1)$. We choose $T = \mu$ and set parameters in Assumption 3 as $A_g^{(1)} \asymp \psi + \mu$, $U_g^{(1)} \asymp (\hat{L} + \mu)\sqrt{n} + \hat{B}$, $P_0^{(1)} \asymp P_0' + \exp(-\Omega(n))$, $A_g^{(2)} \asymp \psi + \mu$, $U_g^{(2)} = 0$, $P_0^{(2)} = 0$. Now we are ready to apply Theorem 1 to this model. We obtain:

**Corollary 3.** *Consider Assumption 1 with $\hat{L}$-Lipschitz $f$, suppose that $\mathbb{P}(f(0) \leq \hat{B}) \geq 1 - P_0'$, and define the parameters $\mu = \mathbb{E}[f(g)g]$, $\psi = \|f(g)\|_{\psi_2}$ with $g \sim \mathcal{N}(0,1)$. Let $\epsilon \in (0,1)$ be any given recovery accuracy. If $m \gtrsim \frac{(\mu+\psi)k}{\epsilon^2} \log\left(\frac{Lr\sqrt{m}[n(\mu+\epsilon)(\hat{L}+\mu)+\sqrt{n}\mu\hat{B}+\psi]}{(\mu+\psi)\epsilon}\right)$, then with probability at least $1 - 2m\exp(-cn) - mP_0' - c_1\exp(-\Omega(k))$ on a single draw of $(\boldsymbol{a}_i, f_i)_{i=1}^m$, we have the uniform signal recovery guarantee $\|\hat{\boldsymbol{x}} - \mu\boldsymbol{x}^*\|_2 \leq \epsilon$ for all $\boldsymbol{x}^* \in \mathcal{K}$, where $\hat{\boldsymbol{x}}$ is the solution to (2.1) with $\boldsymbol{y} = \boldsymbol{f}(\boldsymbol{A}\boldsymbol{x}^*)$ and $T = \mu$.*

**Remark 4.** *While the main result of [33] is non-uniform, it was noted in [33, Section 5] that a similar uniform error rate can be established for any deterministic 1-Lipschitz $f$. Our result here is more general in that the $\hat{L}$-Lipschitz $f$ is possibly random. Note that randomness on $f$ is significant because it provides much more flexibility (e.g., additive random noise).*

**Remark 5.** *For SIM with unknown $f_i$ it may seem impractical to use (2.1) as it requires $\mu = \mathbb{E}[f(g)g]$ where $g \sim \mathcal{N}(0,1)$. However, by assuming $\mu\boldsymbol{x}^* \in \mathcal{K} = G(\mathbb{B}_2^k(r))$ as in [33], which is natural for sufficiently expressive $G(\cdot)$, we can simply use $\boldsymbol{x} \in \mathcal{K}$ as constraint in (2.1). Our Corollary 3 remains valid in this case under some inessential changes of $\log\mu$ factors in the sample complexity.*

**(D) Uniformly quantized GCS with dithering.** The uniform quantizer with resolution $\delta > 0$ is defined as $\mathcal{Q}_\delta(a) = \delta\left(\lfloor\frac{a}{\delta}\rfloor + \frac{1}{2}\right)$ for $a \in \mathbb{R}$. Using dithering $\tau_i \sim \mathcal{U}[-\frac{\delta}{2}, \frac{\delta}{2}]$, we suppose that the observations are $y_i = \mathcal{Q}_\delta(\boldsymbol{a}_i^\top \boldsymbol{x}^* + \tau_i)$. This satisfies Assumption 2 with $(B_0, L_0, \beta_0) = (\delta, 0, \delta)$. We set $T = 1$ and take parameters for Assumption 3 as follows: $A_g^{(1)}, U_g^{(1)}, A_g^{(2)}, U_g^{(2)} \asymp \delta$, and $P_0^{(1)} = P_0^{(2)} = 0$. We take $\beta = \beta_1 \asymp \frac{k\delta}{m}$ to confirm (2.6). With these parameters, we obtain the following from Theorem 1.

**Corollary 4.** *Consider Assumption 1 with $f(\cdot) = \mathcal{Q}_\delta(\cdot + \tau)$, $\tau \sim \mathcal{U}[-\frac{\delta}{2}, \frac{\delta}{2}]$ for some quantization resolution $\delta > 0$. Let $\epsilon > 0$ be any given recovery accuracy. If $m \gtrsim \frac{\delta^2 k}{\epsilon^2} \log\left(\frac{Lr\sqrt{mn}}{\epsilon \wedge [k\delta/(m\sqrt{n})]}\right)$, then with probability at least $1 - 2m\exp(-cn) - c_1\exp(-\Omega(k))$ on a single draw of $(\boldsymbol{a}_i, \tau_i)_{i=1}^m$, we have the uniform recovery guarantee $\|\hat{\boldsymbol{x}} - \boldsymbol{x}^*\|_2 \leq \epsilon$ for all $\boldsymbol{x}^* \in \mathcal{K}$, where $\hat{\boldsymbol{x}}$ is the solution to (2.1) with $\boldsymbol{y} = \mathcal{Q}_\delta(\boldsymbol{A}\boldsymbol{x} + \boldsymbol{\tau})$ and $T = 1$ (here, $\boldsymbol{\tau} = [\tau_1, \ldots, \tau_m]^\top$).*

**Remark 6.** *While this dithered uniform quantized model has been widely studied in traditional CS (e.g., non-uniform recovery [8, 48], uniform recovery [17, 52]), it has not been investigated in GCS even for non-uniform recovery. Thus, this is new to the best of our knowledge.*

A simple extension to the noisy model $\boldsymbol{y} = \boldsymbol{f}(\boldsymbol{A}\boldsymbol{x}^*) + \boldsymbol{\eta}$ where $\boldsymbol{\eta} \in \mathbb{R}^m$ has i.i.d. sub-Gaussian entries can be obtained by a fairly straightforward extension of our analysis; see Appendix F.

## 3 Proof Sketch

To provide a sketch of our proof, we begin with the optimality condition $\|\boldsymbol{y} - \boldsymbol{A}\hat{\boldsymbol{x}}\|_2^2 \leq \|\boldsymbol{y} - \boldsymbol{A}(T\boldsymbol{x}^*)\|_2^2$. We expand the square and plug in $\boldsymbol{y} = \boldsymbol{f}(\boldsymbol{A}\boldsymbol{x}^*)$ to obtain

$$\left\|\frac{\boldsymbol{A}}{\sqrt{m}}(\hat{\boldsymbol{x}} - T\boldsymbol{x}^*)\right\|_2^2 \leq \frac{2}{m}\langle \boldsymbol{f}(\boldsymbol{A}\boldsymbol{x}^*) - T\boldsymbol{A}\boldsymbol{x}^*, \boldsymbol{A}(\hat{\boldsymbol{x}} - T\boldsymbol{x}^*)\rangle. \tag{3.1}$$

For the final goal $\|\hat{\boldsymbol{x}} - T\boldsymbol{x}^*\|_2 \leq \epsilon$, up to rescaling, it is enough to prove $\|\hat{\boldsymbol{x}} - T\boldsymbol{x}^*\|_2 \leq 3\epsilon$. We assume for convenience that $\|\hat{\boldsymbol{x}} - T\boldsymbol{x}^*\|_2 > 2\epsilon$, without loss of generality. Combined with

$\hat{x}, T\boldsymbol{x}^* \in T\mathcal{K}$, we know $\hat{x} - T\boldsymbol{x}^* \in \mathcal{K}_\epsilon^-$, where $\mathcal{K}_\epsilon^- := (T\mathcal{K}^-) \cap (\mathbb{B}_2^n(2\epsilon))^c$, $\mathcal{K}^- = \mathcal{K} - \mathcal{K}$. We further define

$$(\mathcal{K}_\epsilon^-)^* := \{ \boldsymbol{z}/\|\boldsymbol{z}\|_2 : \boldsymbol{z} \in \mathcal{K}_\epsilon^- \} \tag{3.2}$$

where the normalized error lives, i.e. $\frac{\hat{x} - T\boldsymbol{x}^*}{\|\hat{x} - T\boldsymbol{x}^*\|_2} \in (\mathcal{K}_\epsilon^-)^*$. Our strategy is to establish a uniform lower bound (resp., upper bound) for the left-hand side (resp., the right-hand side) of (3.1). We emphasize that these bounds must hold uniformly for all $\boldsymbol{x}^* \in \mathcal{K}$.

It is relatively easy to use set-restricted eigenvalue condition (S-REC) [2] to establish a uniform lower bound for the left-hand side of (3.1), see Appendix B.1 for more details. It is significantly more challenging to derive an upper bound for the right-hand side of (3.1). As the upper bound must hold uniformly for all $\boldsymbol{x}^*$, we first take the supremum over $\boldsymbol{x}^*$ and $\hat{x}$ and consider bounding the following:

$$\begin{aligned}
\mathcal{R} &:= \frac{1}{m} \langle \boldsymbol{f}(A\boldsymbol{x}^*) - TA\boldsymbol{x}^*, A(\hat{x} - T\boldsymbol{x}^*) \rangle \\
&= \frac{1}{m} \sum_{i=1}^m \left( f_i(\boldsymbol{a}_i^\top \boldsymbol{x}^*) - T\boldsymbol{a}_i^\top \boldsymbol{x}^* \right) \cdot \left( \boldsymbol{a}_i^\top [\hat{x} - T\boldsymbol{x}^*] \right) \\
&\leq \|\hat{x} - T\boldsymbol{x}^*\|_2 \cdot \sup_{\boldsymbol{x} \in \mathcal{K}} \sup_{\boldsymbol{v} \in (\mathcal{K}_\epsilon^-)^*} \frac{1}{m} \sum_{i=1}^m \left( f_i(\boldsymbol{a}_i^\top \boldsymbol{x}) - T\boldsymbol{a}_i^\top \boldsymbol{x} \right) \cdot \left( \boldsymbol{a}_i^\top \boldsymbol{v} \right) := \|\hat{x} - T\boldsymbol{x}^*\|_2 \cdot \mathcal{R}_u,
\end{aligned} \tag{3.3}$$

where $(\mathcal{K}_\epsilon^-)^*$ is defined in (3.2). Clearly, $\mathcal{R}_u$ is the supremum of a product process, whose factors are indexed by $\boldsymbol{x} \in \mathcal{K}$ and $\boldsymbol{v} \in (\mathcal{K}_\epsilon^-)^*$. It is, in general, challenging to control a product process, and existing results often require both factors to satisfy a certain "sub-Gaussian increments" condition (e.g., [36, 37]). However, the first factor of $\mathcal{R}_u$ (i.e., $f_i(\boldsymbol{a}_i^\top \boldsymbol{x}^*) - T\boldsymbol{a}_i^\top \boldsymbol{x}^*$) does not admit such a condition when $f_i$ is not continuous (e.g., the 1-bit model $f_i = \mathrm{sign}(\cdot)$). We will construct the Lipschitz approximation of $f_i$ to overcome this difficulty shortly in Section 3.1.

**Remark 7.** *We note that these challenges stem from our pursuit of uniform recovery. In fact, a non-uniform guarantee for SIM was presented in [33, Theorem 1]. In its proof, the key ingredient is [33, Lemma 3] that bounds $\mathcal{R}_u$ without the supremum on $\boldsymbol{x}$. This can be done as long as $f_i(\boldsymbol{a}_i^\top \boldsymbol{x}^*)$ is sub-Gaussian, while the potential discontinuity of $f_i$ is totally unproblematic.*

### 3.1 Lipschitz Approximation

For any $x_0 \in \mathscr{D}_{f_i}$ we define the one-sided limits as $f_i^-(x_0) = \lim_{x \to x_0^-} f_i(x)$ and $f_i^+(x_0) = \lim_{x \to x_0^+} f_i(x)$, and write their average as $f_i^a(x_0) = \frac{1}{2}(f_i^-(x_0) + f_i^+(x_0))$. Given any approximation accuracy $\beta \in (0, \frac{\beta_0}{2})$, we construct the Lipschitz continuous function $f_{i,\beta}$ as:

$$f_{i,\beta}(x) = \begin{cases} f_i(x), & \text{if } x \notin \mathscr{D}_{f_i} + [-\frac{\beta}{2}, \frac{\beta}{2}] \\ f_i^a(x_0) - \frac{2[f_i^a(x_0) - f_i(x_0 - \frac{\beta}{2})](x_0 - x)}{\beta}, & \text{if } \exists x_0 \in \mathscr{D}_{f_i} \text{ s.t. } x \in [x_0 - \frac{\beta}{2}, x_0] \\ f_i^a(x_0) + \frac{2[f_i(x_0 + \frac{\beta}{2}) - f_i^a(x_0)](x - x_0)}{\beta}, & \text{if } \exists x_0 \in \mathscr{D}_{f_i}, \text{ s.t. } x \in [x_0, x_0 + \frac{\beta}{2}] \end{cases} \tag{3.4}$$

We have defined the approximation error $\varepsilon_{i,\beta}(\cdot) = f_{i,\beta}(\cdot) - f_i(\cdot)$ in Assumption 3. An important observation is that both $f_{i,\beta}$ and $|\varepsilon_{i,\beta}|$ are Lipschitz continuous (see Lemma 1 below). Here, it is crucial to consider $|\varepsilon_{i,\beta}|$ rather than $\varepsilon_{i,\beta}$ as the latter is not continuous; see Figure 1 for an intuitive graphical illustration and more explanations in Appendix B.2.

**Lemma 1.** *With $B_0, L_0, \beta_0$ given in Assumption 2, for any $\beta \in (0, \frac{\beta_0}{2})$, $f_{i,\beta}$ is $(L_0 + \frac{B_0}{\beta})$-Lipschitz over $\mathbb{R}$, and $|\varepsilon_{i,\beta}|$ is $(2L_0 + \frac{B_0}{\beta})$-Lipschitz over $\mathbb{R}$.*

### 3.2 Bounding the product process

We now present our technique to bound $\mathcal{R}_u$. Recall that $\xi_{i,\beta}(a)$ and $\varepsilon_{i,\beta}(a)$ were defined in (2.2). By Lemma 1, $\xi_{i,\beta}$ is $(L_0 + T + \frac{B_0}{\beta})$-Lipschitz. Now we use $f_i(a) - Ta = \xi_{i,\beta}(a) - \varepsilon_{i,\beta}$ to decompose $\mathcal{R}_u$ (in the following, we sometimes shorten "$\sup_{\boldsymbol{x} \in \mathcal{K}} \sup_{\boldsymbol{v} \in (\mathcal{K}_\epsilon^-)^*}$" as "$\sup_{\boldsymbol{x}, \boldsymbol{v}}$"):

$$\mathcal{R}_u \leq \underbrace{\sup_{\boldsymbol{x}, \boldsymbol{v}} \frac{1}{m} \sum_{i=1}^m \xi_{i,\beta}(\boldsymbol{a}_i^\top \boldsymbol{x}) \cdot \left( \boldsymbol{a}_i^\top \boldsymbol{v} \right)}_{\mathcal{R}_{u1}} + \underbrace{\sup_{\boldsymbol{x}, \boldsymbol{v}} \frac{1}{m} \sum_{i=1}^m |\varepsilon_{i,\beta}(\boldsymbol{a}_i^\top \boldsymbol{x})| \, |\boldsymbol{a}_i^\top \boldsymbol{v}|}_{\mathcal{R}_{u2}}. \tag{3.5}$$

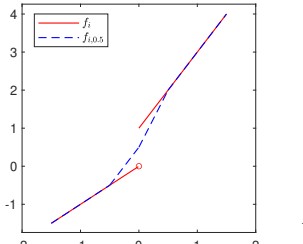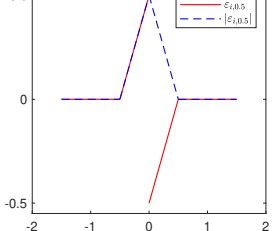

Figure 1: (Left): $f_i$ and its approximation $f_{i,0.5}$; (Right): approximation error $\varepsilon_{i,0.5}, |\varepsilon_{i,0.5}|$.

It remains to control $\mathscr{R}_{u1}$ and $\mathscr{R}_{u2}$. By the Lipschitz continuity of $\xi_{i,\beta}$ and $|\varepsilon_{i,\beta}|$, the factors of $\mathscr{R}_{u1}$ and $\mathscr{R}_{u2}$ admit sub-Gaussian increments, so it is natural to first center them and then invoke the concentration inequality for product process due to Mendelson [36, Theorem 1.13], which we restate in Lemma 5 (Appendix A). However, this does not produce a tight bound and would eventually require $\tilde{O}(k/\epsilon^4)$ to achieve a uniform $\ell_2$-error of $\epsilon$, as is the case in [17, Section 4].

In fact, Lemma 5 is based on *Gaussian width* and hence blind to the fact that $\mathcal{K}, (\mathcal{K}_\epsilon^-)^*$ here have low *metric entropy* (Lemma 6). By characterizing the low intrinsic dimension of index sets via metric entropy, we develop the following concentration inequality that can produce tighter bound for $\mathscr{R}_{u1}$ and $\mathscr{R}_{u2}$. This also allows us to derive uniform error rates sharper than those in [17, Section 4].

**Theorem 2.** *Let $g_{\boldsymbol{x}} = g_{\boldsymbol{x}}(\boldsymbol{a})$ and $h_{\boldsymbol{v}} = h_{\boldsymbol{v}}(\boldsymbol{a})$ be stochastic processes indexed by $\boldsymbol{x} \in \mathcal{X} \subset \mathbb{R}^{p_1}, \boldsymbol{v} \in \mathcal{V} \subset \mathbb{R}^{p_2}$, both defined with respect to a common random variable $\boldsymbol{a}$. Assume that:*

- *(A1.) $g_{\boldsymbol{x}}(\boldsymbol{a}), h_{\boldsymbol{v}}(\boldsymbol{a})$ are sub-Gaussian for some $(A_g, A_h)$ and admit sub-Gaussian increments regarding $\ell_2$ distance for some $(M_g, M_h)$:*

$$\|g_{\boldsymbol{x}}(\boldsymbol{a}) - g_{\boldsymbol{x}'}(\boldsymbol{a})\|_{\psi_2} \le M_g \|\boldsymbol{x} - \boldsymbol{x}'\|_2, \ \|g_{\boldsymbol{x}}(\boldsymbol{a})\|_{\psi_2} \le A_g, \ \forall\, \boldsymbol{x}, \boldsymbol{x}' \in \mathcal{X};$$
$$\|h_{\boldsymbol{v}}(\boldsymbol{a}) - h_{\boldsymbol{v}'}(\boldsymbol{a})\|_{\psi_2} \le M_h \|\boldsymbol{v} - \boldsymbol{v}'\|_2, \ \|h_{\boldsymbol{v}}(\boldsymbol{a})\|_{\psi_2} \le A_h, \ \forall\, \boldsymbol{v}, \boldsymbol{v}' \in \mathcal{V}. \quad (3.6)$$

- *(A2.) On a single draw of $\boldsymbol{a}$, for some $(L_g, U_g, L_h, U_h)$ the following events simultaneously hold with probability at least $1 - P_0$:*

$$|g_{\boldsymbol{x}}(\boldsymbol{a}) - g_{\boldsymbol{x}'}(\boldsymbol{a})| \le L_g \|\boldsymbol{x} - \boldsymbol{x}'\|_2, \ |g_{\boldsymbol{x}}(\boldsymbol{a})| \le U_g, \ \forall\, \boldsymbol{x}, \boldsymbol{x}' \in \mathcal{X};$$
$$|h_{\boldsymbol{v}}(\boldsymbol{a}) - h_{\boldsymbol{v}'}(\boldsymbol{a})| \le L_h \|\boldsymbol{v} - \boldsymbol{v}'\|_2, \ |h_{\boldsymbol{v}}(\boldsymbol{a})| \le U_h, \ \forall\, \boldsymbol{v}, \boldsymbol{v}' \in \mathcal{V}. \quad (3.7)$$

*Let $\boldsymbol{a}_1, ..., \boldsymbol{a}_m$ be i.i.d. copies of $\boldsymbol{a}$, and introduce the shorthand $S_{g,h} = L_g U_h + M_g A_h$ and $T_{g,h} = L_h U_g + M_h A_g$. If $m \gtrsim \mathscr{H}\left(\mathcal{X}, \frac{A_g A_h}{\sqrt{m} S_{g,h}}\right) + \mathscr{H}\left(\mathcal{V}, \frac{A_g A_h}{\sqrt{m} T_{g,h}}\right)$, where $\mathscr{H}(\cdot, \cdot)$ is the metric entropy defined in Definition 2, then with probability at least $1 - mP_0 - 2\exp\big[ -\Omega\big(\mathscr{H}(\mathcal{X}, \frac{A_g A_h}{\sqrt{m} S_{g,h}}) + \mathscr{H}(\mathcal{V}, \frac{A_g A_h}{\sqrt{m} T_{g,h}})\big)\big]$ we have $I \lesssim \frac{A_g A_h}{\sqrt{m}}\sqrt{\mathscr{H}(\mathcal{X}, \frac{A_g A_h}{\sqrt{m} S_{g,h}}) + \mathscr{H}(\mathcal{V}, \frac{A_g A_h}{\sqrt{m} T_{g,h}})}$, where $I := \sup_{\boldsymbol{x} \in \mathcal{X}} \sup_{\boldsymbol{v} \in \mathcal{V}} \left|\frac{1}{m}\sum_{i=1}^m \big(g_{\boldsymbol{x}}(\boldsymbol{a}_i) h_{\boldsymbol{v}}(\boldsymbol{a}_i) - \mathbb{E}[g_{\boldsymbol{x}}(\boldsymbol{a}_i) h_{\boldsymbol{v}}(\boldsymbol{a}_i)]\big)\right|$ is the supremum of a product process.*

**Remark 8.** *We use $\mathscr{R}_{u2}$ as an example to illustrate the advantage of Theorem 2 over Lemma 5. The key step is on bounding the centered process*

$$\mathscr{R}_{u2,c} := \sup_{\boldsymbol{x} \in \mathcal{K}} \sup_{\boldsymbol{v} \in (\mathcal{K}_\epsilon^-)^*} \big\{|\varepsilon_{i,\beta}(\boldsymbol{a}_i^\top \boldsymbol{x})||\boldsymbol{a}_i^\top \boldsymbol{v}| - \mathbb{E}[|\varepsilon_{i,\beta}(\boldsymbol{a}_i^\top \boldsymbol{x})||\boldsymbol{a}_i^\top \boldsymbol{v}|]\big\}.$$

*Let $g_{\boldsymbol{x}}(\boldsymbol{a}_i) = |\varepsilon_{i,\beta}(\boldsymbol{a}_i^\top \boldsymbol{x})|$ and $h_{\boldsymbol{v}}(\boldsymbol{a}_i) = |\boldsymbol{a}_i^\top \boldsymbol{v}|$, then one can use Theorem 2 or Lemma 5 to bound $\mathscr{R}_{u2,c}$. Note that $\|\boldsymbol{a}_i^\top \boldsymbol{v}\|_{\psi_2} = O(1)$ justifies the choice $A_h = O(1)$, and both $\mathscr{H}(\mathcal{K}, \eta)$ and $\mathscr{H}((\mathcal{K}_\epsilon^-)^*, \eta)$ depend linearly on $k$ but only logarithmically on $\eta$ (Lemma 6), so Theorem 2 could bound $\mathscr{R}_{u2,c}$ by $\tilde{O}\big(A_g \sqrt{k/m}\big)$ that depends on $M_g$ in a logarithmic manner. However, the bound produced by Lemma 5 depends linearly on $M_g$; see term $\frac{M_g A_h \omega(\mathcal{K})}{\sqrt{m}}$ in (A.1). From (3.6), $M_g$ should be proportional to the Lipschitz constant of $|\varepsilon_{i,\beta}|$, which scales as $\frac{1}{\beta}$ (Lemma 1). The issue is that in many cases we need to take extremely small $\beta$ to guarantee that (2.6) holds true (e.g., we take $\beta \asymp k/m$ in 1-bit GCS). Thus, Lemma 5 produces a worse bound compared to our Theorem 2.*

## 4 Conclusion

In this work, we built a unified framework for uniform signal recovery in nonlinear generative compressed sensing. We showed that using generalized Lasso, a sample size of $\tilde{O}(k/\epsilon^2)$ suffices to uniformly recover all $\boldsymbol{x} \in G(\mathbb{B}_2^k(r))$ up to an $\ell_2$-error of $\epsilon$. We specialized our main theorem to 1-bit GCS with/without dithering, single index model, and uniformly quantized GCS, deriving uniform guarantees that are new or exhibit some advantages over existing ones. Unlike [33], our proof is free of any non-trivial embedding property. As part of our technical contributions, we constructed the Lipschitz approximation to handle potential discontinuity in the observation model, and also developed a concentration inequality to derive tighter bound for the product processes arising in the proof, allowing us to obtain a uniform error rate faster than [17]. Possible future directions include extending our framework to handle the adversarial noise and representation error.

**Acknowledgment.** J. Chen was supported by a Hong Kong PhD Fellowship from the Hong Kong Research Grants Council (RGC). J. Scarlett was supported by the Singapore National Research Foundation (NRF) under grant A-0008064-00-00. M. K. Ng was partially supported by the HKRGC GRF 17201020, 17300021, CRF C7004-21GF and Joint NSFC-RGC N-HKU76921.

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

# Supplementary Material

## A Unified Framework for Uniform Signal Recovery in Nonlinear Generative Compressed Sensing

### (NeurIPS 2023)

## A   Technical Lemmas

**Lemma 2.** (Lemma 2.7.7, [50]). *Let $X, Y$ be sub-Gaussian, then $XY$ is sub-exponential with $\|XY\|_{\psi_1} \leq \|X\|_{\psi_2}\|Y\|_{\psi_2}$.*

**Lemma 3.** (Centering, [50, Exercise 2.7.10]). *For some absolute constant $C$, $\|X - \mathbb{E}X\|_{\psi_1} \leq C\|X\|_{\psi_1}$.*

**Lemma 4.** (Bernstein's inequality, [50, Theorem 2.8.1]). *Let $X_1, ..., X_N$ be independent, zero-mean, sub-exponential random variables. Then for every $t \geq 0$, for some absolute constant $c$ we have*

$$\mathbb{P}\left(\left|\sum_{i=1}^{N} X_i\right| \geq t\right) \leq 2\exp\left(-c\min\left\{\frac{t^2}{\sum_{i=1}^{N}\|X_i\|_{\psi_1}^2}, \frac{t}{\max_{1\leq i\leq N}\|X_i\|_{\psi_1}}\right\}\right)$$

**Lemma 5.** ( [36], statement adapted from [16, Theorem 8]). *Let $g_{\boldsymbol{x}} = g_{\boldsymbol{x}}(\boldsymbol{a})$ and $h_{\boldsymbol{v}} = h_{\boldsymbol{v}}(\boldsymbol{a})$ be stochastic processes indexed by $\boldsymbol{x} \in \mathcal{X} \subset \mathbb{R}^{p_1}$, $\boldsymbol{v} \in \mathcal{V} \subset \mathbb{R}^{p_2}$, both defined on some common random variable $\boldsymbol{a}$. Assume that (A1.) in Theorem 2 holds, and let $\boldsymbol{a}_1, ..., \boldsymbol{a}_m$ be i.i.d. copies of $\boldsymbol{a}$. Then for any $u \geq 1$, with probability at least $1 - 2\exp(-cu^2)$ we have the bound*

$$\sup_{\substack{\boldsymbol{x}\in\mathcal{X} \\ \boldsymbol{v}\in\mathcal{V}}}\left|\frac{1}{m}\sum_{i=1}^{m}\left(g_{\boldsymbol{x}}(\boldsymbol{a}_i)h_{\boldsymbol{v}}(\boldsymbol{a}_i) - \mathbb{E}[g_{\boldsymbol{x}}(\boldsymbol{a}_i)h_{\boldsymbol{v}}(\boldsymbol{a}_i)]\right)\right|$$

$$\leq C\Big(\frac{(M_g\cdot\omega(\mathcal{X}) + u\cdot A_g)\cdot(M_h\cdot\omega(\mathcal{V}) + u\cdot A_h)}{m} \tag{A.1}$$

$$+ \frac{A_g\cdot M_h\cdot\omega(\mathcal{V}) + A_h\cdot M_g\cdot\omega(\mathcal{X}) + u\cdot A_g A_h}{\sqrt{m}}\Big),$$

*where $\omega(\cdot)$ is the Gaussian width defined as $\omega(\mathcal{X}) = \mathbb{E}\sup_{\boldsymbol{x}\in\mathcal{X}}\boldsymbol{g}^\top\boldsymbol{x}$ where $\boldsymbol{g} \sim \mathcal{N}(0, \boldsymbol{I}_{p_1})$.*

The proofs of the remaining lemmas will be provided in Appendix D. (Some simple facts such as Lemma 8 were already used in prior works; while we provide the proofs for completeness.)

**Lemma 6.** (Metric entropy of some constraint sets). *Assume $\mathcal{K} = G(\mathbb{B}_2^k(r))$ for some $L$-Lipschitz generative model $G$. Let $\mathcal{K}^- = \mathcal{K} - \mathcal{K}$, for some $T > 0, \epsilon \in (0, 1)$ let $\mathcal{K}_\epsilon^- := (T\mathcal{K}^-) \cap \left(\mathbb{B}_2^n(2\epsilon)\right)^c$, and further define $(\mathcal{K}_\epsilon^-)^* = \left\{\frac{\boldsymbol{z}}{\|\boldsymbol{z}\|_2} : \boldsymbol{z} \in \mathcal{K}_\epsilon^-\right\}$. Then for any $\eta \in (0, Lr)$, we have*

$$\mathscr{H}(\mathcal{K}, \eta) \leq k\log\frac{3Lr}{\eta}, \ \mathscr{H}(\mathcal{K}^-, \eta) \leq 2k\log\frac{6Lr}{\eta},$$

$$\mathscr{H}(\mathcal{K}_\epsilon^-, \eta) \leq 2k\log\frac{12TLr}{\eta}, \ \mathscr{H}\left((\mathcal{K}_\epsilon^-)^*, \eta\right) \leq 2k\log\frac{12TLr}{\epsilon\eta},$$

*where $\mathscr{H}(\cdot, \cdot)$ is the metric entropy defined in Definition 2.*

**Lemma 7.** (Bound the $\ell_2$-norm of Gaussian vector). *If $\boldsymbol{a} \sim \mathcal{N}(0, \boldsymbol{I}_n)$, then $\mathbb{P}\left(|\|\boldsymbol{a}\|_2 - \sqrt{n}| \geq t\right) \leq 2\exp(-Ct^2)$. In particular, setting $t \asymp \sqrt{n}$ yields $\mathbb{P}(\|\boldsymbol{a}\|_2 \geq \sqrt{n}) \leq 2\exp(-\Omega(n))$.*

In the following, Lemmas 8-11 indicate suitable choices of $T$ in the concrete models we consider. These choices can make $\rho(\boldsymbol{x})$ in (2.3) sufficiently small or even zero.

**Lemma 8.** (Choice of $T$ in 1-bit GCS). *If $\boldsymbol{a} \sim \mathcal{N}(0, \boldsymbol{I}_n)$, then for any $\boldsymbol{x} \in \mathbb{S}^{n-1}$ it holds that $\mathbb{E}[\text{sign}(\boldsymbol{a}^\top\boldsymbol{x})\boldsymbol{a}] = \sqrt{\frac{2}{\pi}}\boldsymbol{x}$.*

**Lemma 9.** (Choice of $T$ in 1-bit GCS with dithering). *If $\boldsymbol{a} \sim \mathcal{N}(0, \boldsymbol{I}_n)$ and $\tau \sim \mathscr{U}[-\lambda, \lambda]$ are independent, and $\lambda = CR\sqrt{\log m}$ with sufficiently large $C$, then for any $\boldsymbol{x} \in \mathbb{B}_2^n(R)$ it holds that $\|\mathbb{E}[\text{sign}(\boldsymbol{a}^\top\boldsymbol{x} + \tau)\boldsymbol{a}] - \frac{\boldsymbol{x}}{\lambda}\|_2 = O(m^{-9})$.*

**Lemma 10.** (Choice of $T$ in SIM). *If $\boldsymbol{a} \sim \mathcal{N}(0, \boldsymbol{I}_n)$, for some function $f$ and any $\boldsymbol{x} \in \mathbb{S}^{n-1}$ it holds that $\mathbb{E}[f(\boldsymbol{a}^\top \boldsymbol{x})\boldsymbol{a}] = \mu \boldsymbol{x}$ for $\mu = \mathbb{E}[f(g)g]$ with $g \sim \mathcal{N}(0, 1)$.*

**Lemma 11.** (Choice of $T$ in uniformly quantized GCS with dithering). *Given any $\delta > 0$, let $\tau \sim \mathscr{U}[-\frac{\delta}{2}, \frac{\delta}{2}]$ and $\mathcal{Q}_\delta(\cdot) = \delta(\lfloor \frac{\cdot}{\delta} \rfloor + \frac{1}{2})$. Then, for any $a \in \mathbb{R}$, it holds that $\mathbb{E}[\mathcal{Q}_\delta(a + \tau)] = a$. In particular, let $\boldsymbol{a} \in \mathbb{R}^n$ be a random vector satisfying $\mathbb{E}(\boldsymbol{a}\boldsymbol{a}^\top) = \boldsymbol{I}_n$, and $\tau \sim \mathscr{U}[-\frac{\delta}{2}, \frac{\delta}{2}]$ be independent of $\boldsymbol{a}$, then we have $\mathbb{E}[\mathcal{Q}_\delta(\boldsymbol{a}^\top \boldsymbol{x} + \tau)\boldsymbol{a}] = \boldsymbol{x}$.*

Lemma 12 facilitates our analysis of the uniform quantizer.

**Lemma 12.** *Let $f_i(\cdot) = \delta(\lfloor \frac{\cdot + \tau_i}{\delta} \rfloor + \frac{1}{2})$ for $\tau_i \sim \mathscr{U}[-\frac{\delta}{2}, \frac{\delta}{2}]$, and $f_{i,\beta}(\cdot)$ be defined in (3.4) for some $0 < \beta < \frac{\delta}{2}$. Moreover, let $\xi_{i,\beta}(a) = f_{i,\beta}(a) - a$, $\varepsilon_{i,\beta}(a) = f_{i,\beta}(a) - f_i(a)$, then for any $a \in \mathbb{R}$, $|\xi_{i,\beta}(a)| \le 2\delta$, $|\varepsilon_{i,\beta}(a)| \le \delta$ holds deterministically.*

More generally, the approximation error $|\varepsilon_{i,\beta}(a)|$ can always be bounded as follows.

**Lemma 13.** *Suppose that $f_i$ satisfies Assumption 2, and for any $\beta \in [0, \frac{\beta_0}{2}]$ we construct $f_{i,\beta}$ as in (3.4). Then, for any $a \in \mathbb{R}$, we have $|\varepsilon_{i,\beta}(a)| \le \left(\frac{3L_0\beta}{2} + B_0\right)\mathbb{1}(a \in \mathscr{D}_{f_i} + [-\frac{\beta}{2}, \frac{\beta}{2}])$.*

# B   More Details of the Proof Sketch

## B.1   Set-Restricted Eigenvalue Condition

**Definition 4.** *Let $\mathcal{S} \subset \mathbb{R}^n$. For parameters $\gamma, \delta > 0$, a matrix $\boldsymbol{A} \in \mathbb{R}^{m \times n}$ is said to satisfy S-REC($\mathcal{S}, \gamma, \delta$) if the following holds:*

$$\|\boldsymbol{A}(\boldsymbol{x}_1 - \boldsymbol{x}_2)\|_2 \ge \gamma \|\boldsymbol{x}_1 - \boldsymbol{x}_2\|_2 - \delta, \ \forall \ \boldsymbol{x}_1, \boldsymbol{x}_2 \in \mathcal{S}.$$

It was proved in [2] that $\frac{1}{\sqrt{m}}\boldsymbol{A}$ satisfies the S-REC with high probability if the entries of $\boldsymbol{A}$ are i.i.d. standard Gaussian.

**Lemma 14.** (Lemma 4.1 in [2]). *Let $G : \mathbb{B}_2^k(r) \to \mathbb{R}^n$ be $L$-Lipschitz for some $r, L > 0$, and define $\mathcal{K} = G(\mathbb{B}_2^k(r))$. For any $\alpha \in (0, 1)$, if $\boldsymbol{A} \in \mathbb{R}^{m \times n}$ has i.i.d. $\mathcal{N}(0, 1)$ entries, and $m = \Omega\left(\frac{k}{\alpha^2} \log \frac{Lr}{\delta}\right)$, then $\frac{1}{\sqrt{m}}\boldsymbol{A}$ satisfies S-REC($\mathcal{K}, 1 - \alpha, \delta$) with probability at least $1 - \exp(-\Omega(\alpha^2 m))$.*

## B.2   Lipschitz Approximation

The approximation error $\varepsilon_{i,\beta}(\cdot)$ can be expanded as:

$$\varepsilon_{i,\beta}(x) = \begin{cases} 0 & , \quad \text{if } x \notin \mathscr{D}_{f_i} + [-\frac{\beta}{2}, \frac{\beta}{2}] \\ f_i^a(x_0) - f_i(x) - \frac{2[f_i^a(x_0) - f_i(x_0 - \frac{\beta}{2})](x_0 - x)}{\beta}, & \text{if } x \in [x_0 - \frac{\beta}{2}, x_0], \ (\exists x_0 \in \mathscr{D}_{f_i}) \\ f_i^a(x_0) - f_i(x) + \frac{2[f_i(x_0 + \frac{\beta}{2}) - f_i^a(x_0)](x - x_0)}{\beta}, & \text{if } x \in [x_0, x_0 + \frac{\beta}{2}], \ (\exists x_0 \in \mathscr{D}_{f_i}) \end{cases}.$$

Although $|\varepsilon_{i,\beta}|$ is Lipschitz continuous, $\varepsilon_{i,\beta}$ is not. In particular, given $x_0 \in \mathscr{D}_{f_i}$ we note that

$$\varepsilon_{i,\beta}^-(x_0) = \lim_{x \to x_0^-} \varepsilon_{i,\beta}(x) = f_i^a(x_0) - f_i^-(x_0) = \frac{1}{2}\left(f_i^+(x_0) - f_i^-(x_0)\right),$$

$$\varepsilon_{i,\beta}^+(x_0) = \lim_{x \to x_0^+} \varepsilon_{i,\beta}(x) = f_i^a(x_0) - f_i^+(x_0) = \frac{1}{2}\left(f_i^-(x_0) - f_i^+(x_0)\right).$$

Thus, it is crucial to include the absolute value for rendering the continuity.

# C   Proofs of Main Results

## C.1   Proof of Theorem 1

*Proof.* Up to rescaling, we only need to prove that $\|\hat{\boldsymbol{x}} - T\boldsymbol{x}^*\|_2 \le 3\epsilon$ holds uniformly for all $\boldsymbol{x}^* \in \mathcal{K}$. We can assume $\|\hat{\boldsymbol{x}} - T\boldsymbol{x}^*\|_2 \ge 2\epsilon$; otherwise, the desired bound is immediate.

**(1) Lower bounding the left-hand side of (3.1).**

We use S-REC to find a lower bound for $\left\|\frac{\boldsymbol{A}}{\sqrt{m}}(\hat{\boldsymbol{x}} - T\boldsymbol{x}^*)\right\|_2^2$. Specifically, we invoke Lemma 14 with $\alpha = \frac{1}{2}$ and $\delta = \frac{\epsilon}{2T}$, which gives that under $m = \Omega\big(k\log\frac{LTr}{\epsilon}\big)$, with probability at least $1 - \exp(-cm)$, the following holds:

$$\left\|\frac{\boldsymbol{A}}{\sqrt{m}}(\boldsymbol{x}_1 - \boldsymbol{x}_2)\right\|_2 \geq \frac{1}{2}\|\boldsymbol{x}_1 - \boldsymbol{x}_2\|_2 - \frac{\epsilon}{2T}, \ \forall\, \boldsymbol{x}_1, \boldsymbol{x}_2 \in \mathcal{K}. \tag{C.1}$$

Recall that we assume $\|\hat{\boldsymbol{x}} - T\boldsymbol{x}^*\|_2 \geq 2\epsilon$, $T^{-1}\hat{\boldsymbol{x}} \in \mathcal{K}$, and $\boldsymbol{x}^* \in \mathcal{K}$, so we set $\boldsymbol{x}_1 = \frac{\hat{\boldsymbol{x}}}{T}, \boldsymbol{x}_2 = \boldsymbol{x}^*$ in (C.1) to obtain

$$\left\|\frac{\boldsymbol{A}}{\sqrt{m}}\Big(\frac{\hat{\boldsymbol{x}}}{T} - \boldsymbol{x}^*\Big)\right\|_2 \geq \frac{1}{2}\left\|\frac{\hat{\boldsymbol{x}}}{T} - \boldsymbol{x}^*\right\|_2 - \frac{\epsilon}{2T} \geq \frac{1}{4T}\|\hat{\boldsymbol{x}} - T\boldsymbol{x}^*\|_2.$$

Thus, the left-hand side of (3.1) can be lower bounded by $\Omega\big(\|\hat{\boldsymbol{x}} - T\boldsymbol{x}^*\|_2^2\big)$.

**(2) Upper bounding the right-hand side of (3.1).**

As analysed in (3.3) and (3.5), the right-hand side of (3.1) is bounded by $2\|\hat{\boldsymbol{x}} - T\boldsymbol{x}^*\|_2 \cdot \big(\mathscr{R}_{u1} + \mathscr{R}_{u2}\big)$, so all that remains is to bound $\mathscr{R}_{u1}, \mathscr{R}_{u2}$. In the rest of the proof, we simply write $\sup_{\boldsymbol{x},\boldsymbol{v}} :=$ $\sup_{\boldsymbol{x}\in\mathcal{K},\boldsymbol{v}\in(\mathcal{K}_\epsilon^-)^*}$ and recall the shorthand $\xi_{i,\beta}(a) = f_{i,\beta}(a) - Ta$. Thus, the first factor of $\mathscr{R}_{u1}$ is given by $\xi_{i,\beta}(\boldsymbol{a}_i^\top \boldsymbol{x})$. By centering, we have

$$\mathscr{R}_{u1} \leq \underbrace{\sup_{\boldsymbol{x},\boldsymbol{v}} \frac{1}{m}\sum_{i=1}^m \big\{[\xi_{i,\beta}(\boldsymbol{a}_i^\top\boldsymbol{x})](\boldsymbol{a}_i^\top\boldsymbol{v}) - \mathbb{E}\big[[\xi_{i,\beta}(\boldsymbol{a}_i^\top\boldsymbol{x})](\boldsymbol{a}_i^\top\boldsymbol{v})\big]\big\}}_{\mathscr{R}_{u1,c}} + \underbrace{\sup_{\boldsymbol{x},\boldsymbol{v}} \mathbb{E}\big[[\xi_{i,\beta}(\boldsymbol{a}_i^\top\boldsymbol{x})](\boldsymbol{a}_i^\top\boldsymbol{v})\big]}_{\mathscr{R}_{u1,e}},$$
$$\tag{C.2}$$

and

$$\mathscr{R}_{u2} \leq \underbrace{\sup_{\boldsymbol{x},\boldsymbol{v}}\left\{\frac{1}{m}\sum_{i=1}^m \big|\varepsilon_{i,\beta}(\boldsymbol{a}_i^\top\boldsymbol{x})\big|\big|\boldsymbol{a}_i^\top\boldsymbol{v}\big| - \mathbb{E}\big[\big|\varepsilon_{i,\beta}(\boldsymbol{a}_i^\top\boldsymbol{x})\big|\big|\boldsymbol{a}_i^\top\boldsymbol{v}\big|\big]\right\}}_{\mathscr{R}_{u2,c}} + \underbrace{\sup_{\boldsymbol{x},\boldsymbol{v}} \mathbb{E}\big[\big|\varepsilon_{i,\beta}(\boldsymbol{a}_i^\top\boldsymbol{x})\big|\big|\boldsymbol{a}_i^\top\boldsymbol{v}\big|\big]}_{\mathscr{R}_{u2,e}}.$$
$$\tag{C.3}$$

We will invoke Theorem 2 multiple times to derive the required bounds.

**(2.1) Bounding the centered product process $\mathscr{R}_{u1,c}$.**

We let $g_{\boldsymbol{x}}(\boldsymbol{a}_i) = \xi_{i,\beta}(\boldsymbol{a}_i^\top\boldsymbol{x})$ and $h_{\boldsymbol{v}}(\boldsymbol{a}_i) = \boldsymbol{a}_i^\top\boldsymbol{v}$, and write

$$\mathscr{R}_{u1,c} = \sup_{\boldsymbol{x},\boldsymbol{v}} \frac{1}{m}\sum_{i=1}^m \big\{g_{\boldsymbol{x}}(\boldsymbol{a}_i)h_{\boldsymbol{v}}(\boldsymbol{a}_i) - \mathbb{E}[g_{\boldsymbol{x}}(\boldsymbol{a}_i)h_{\boldsymbol{v}}(\boldsymbol{a}_i)]\big\}.$$

We verify conditions in Theorem 2 as follows:

- For any $\boldsymbol{x}, \boldsymbol{x}' \in \mathcal{K}$, because $\xi_{i,\beta}$ is $\big(L_0 + \frac{B_0}{\beta} + T\big)$-Lipschitz continuous (Lemma 1), we have

$$\|g_{\boldsymbol{x}}(\boldsymbol{a}_i) - g_{\boldsymbol{x}'}(\boldsymbol{a}_i)\|_{\psi_2} \leq (L_0 + T + \frac{B_0}{\beta})\|\boldsymbol{a}_i^\top\boldsymbol{x} - \boldsymbol{a}_i^\top\boldsymbol{x}'\|_{\psi_2}$$

$$= O\big(L_0 + T + \frac{B_0}{\beta}\big)\|\boldsymbol{x} - \boldsymbol{x}'\|_2.$$

- Since $\boldsymbol{a}_i \sim \mathcal{N}(\boldsymbol{0}, \boldsymbol{I}_n)$, by Lemma 7, with probability $1 - 2\exp(-\Omega(n))$ we have $\|\boldsymbol{a}_i\|_2 = O(\sqrt{n})$. On this event, we have

$$|g_{\boldsymbol{x}}(\boldsymbol{a}_i) - g_{\boldsymbol{x}'}(\boldsymbol{a}_i)| \leq (L_0 + T + \frac{B_0}{\beta})|\boldsymbol{a}_i^\top\boldsymbol{x} - \boldsymbol{a}_i^\top\boldsymbol{x}'|$$

$$\leq (L_0 + T + \frac{B_0}{\beta})\|\boldsymbol{a}_i\|_2\|\boldsymbol{x} - \boldsymbol{x}'\|_2$$

$$= O\Big(\sqrt{n}\Big[L_0 + T + \frac{B_0}{\beta}\Big]\Big)\|\boldsymbol{x} - \boldsymbol{x}'\|_2.$$

- Recall $(\mathcal{K}_\epsilon^-)^*$ in (3.2). Since $(\mathcal{K}_\epsilon^-)^* \subset \mathbb{S}^{n-1}$, for any $\boldsymbol{v}, \boldsymbol{v}' \in (\mathcal{K}_\epsilon^-)^*$, we have $\|\boldsymbol{a}_i^\top \boldsymbol{v} - \boldsymbol{a}_i^\top \boldsymbol{v}'\|_{\psi_2} = O(1)\|\boldsymbol{v} - \boldsymbol{v}'\|_2$, and when $\|\boldsymbol{a}_i\| = O(\sqrt{n})$ we have $|\boldsymbol{a}_i^\top \boldsymbol{v} - \boldsymbol{a}_i^\top \boldsymbol{v}'| \le \|\boldsymbol{a}_i\|_2 \|\boldsymbol{v} - \boldsymbol{v}'\|_2 = O(\sqrt{n})\|\boldsymbol{v} - \boldsymbol{v}'\|_2$. Moreover, because $(\mathcal{K}_\epsilon^-)^* \subset \mathbb{B}_2^n$, for any $\boldsymbol{v} \in (\mathcal{K}_\epsilon^-)^*$ we have $\|\boldsymbol{a}_i^\top \boldsymbol{v}\|_{\psi_2} = O(1)$, $|\boldsymbol{a}_i^\top \boldsymbol{v}| \le \|\boldsymbol{a}_i\|_2 = O(\sqrt{n})$.

Combined with Assumption 3 and its parameters $(A_g^{(1)}, U_g^{(1)}, P_0^{(1)})$ and $(A_g^{(2)}, U_g^{(2)}, P_0^{(2)})$, $\mathscr{R}_{u1,c}$ satisfies the conditions of Theorem 2 with the following parameters

$$M_g \asymp L_0 + T + \frac{B_0}{\beta}, \ A_g = A_g^{(1)}, \ M_h \asymp 1, \ A_h \asymp 1$$

$$L_g \asymp \sqrt{n}\big(L_0 + T + \frac{B_0}{\beta}\big), \ U_g = U_g^{(1)}, \ L_h \asymp \sqrt{n}, \ U_h \asymp \sqrt{n}$$

and $P_0 = P_0^{(1)} + 2\exp(-\Omega(n))$. Now suppose that we have

$$m \gtrsim \mathscr{H}\left(\mathcal{K}, \frac{A_g^{(1)}}{\sqrt{mn}[L_0 + T + \frac{B_0}{\beta}]}\right) + \mathscr{H}\left((\mathcal{K}_\epsilon^-)^*, \frac{A_g^{(1)}}{\sqrt{m}(\sqrt{n}U_g^{(1)} + A_g^{(1)})}\right), \quad \text{(C.4)}$$

and note that by using Lemma 6, (C.4) can be guaranteed by

$$m \gtrsim k \log\left(\frac{Lr\sqrt{m}}{A_g^{(1)}}\Big[n(L_0 + T + \frac{B_0}{\beta}) + \frac{T(\sqrt{n}U_g^{(1)} + A_g^{(1)})}{\epsilon}\Big]\right). \quad \text{(C.5)}$$

Then Theorem 2 yields that the following bound holds with probability at least $1 - mP_0^{(1)} - C\exp(-\Omega(k)) - m\exp(-\Omega(n))$:

$$
\begin{aligned}
|\mathscr{R}_{u1,c}| &\lesssim \frac{A_g^1}{\sqrt{m}} \sqrt{\mathscr{H}\left(\mathcal{K}, \frac{A_g^{(1)}}{\sqrt{mn}[L_0 + T + \frac{B_0}{\beta}]}\right) + \mathscr{H}\left((\mathcal{K}_\epsilon^-)^*, \frac{A_g^{(1)}}{\sqrt{m}(\sqrt{n}U_g^{(1)} + A_g^{(1)})}\right)} \\
&\lesssim A_g^{(1)} \sqrt{\frac{k}{m} \log\left(\frac{Lr\sqrt{m}}{A_g^{(1)}}\Big[n(L_0 + T + \frac{B_0}{\beta}) + \frac{T(\sqrt{n}U_g^{(1)} + A_g^{(1)})}{\epsilon}\Big]\right)}.
\end{aligned}
\quad \text{(C.6)}
$$

**(2.2) Bounding the centered product process $\mathscr{R}_{u2,c}$.**

We let $g_{\boldsymbol{x}}(\boldsymbol{a}_i) = |\varepsilon_{i,\beta}(\boldsymbol{a}_i^\top \boldsymbol{x})|$ and $h_{\boldsymbol{v}}(\boldsymbol{a}_i) = |\boldsymbol{a}_i^\top \boldsymbol{v}|$, and write

$$\mathscr{R}_{u2,c} = \sup_{\boldsymbol{x}, \boldsymbol{v}} \frac{1}{m} \sum_{i=1}^m \big\{g_{\boldsymbol{x}}(\boldsymbol{a}_i)h_{\boldsymbol{v}}(\boldsymbol{a}_i) - \mathbb{E}[g_{\boldsymbol{x}}(\boldsymbol{a}_i)h_{\boldsymbol{v}}(\boldsymbol{a}_i)]\big\}.$$

We verify the conditions in Theorem 2 as follows:

- For any $\boldsymbol{x}, \boldsymbol{x}' \in \mathcal{K}$, because $|\varepsilon_{i,\beta}|$ is $(2L_0 + \frac{B_0}{\beta})$-Lipschitz continuous (Lemma 1), we have

$$\|g_{\boldsymbol{x}}(\boldsymbol{a}_i) - g_{\boldsymbol{x}'}(\boldsymbol{a}_i)\|_{\psi_2} \le (2L_0 + \frac{B_0}{\beta})\|\boldsymbol{a}_i^\top \boldsymbol{x} - \boldsymbol{a}_i^\top \boldsymbol{x}'\|_{\psi_2}$$

$$= O\big(L_0 + \frac{B_0}{\beta}\big)\|\boldsymbol{x} - \boldsymbol{x}'\|_2.$$

- By Lemma 7, with probability at least $1 - 2\exp(-\Omega(n))$ we have $\|\boldsymbol{a}_i\|_2 = O(\sqrt{n})$. On this event, we have

$$|g_{\boldsymbol{x}}(\boldsymbol{a}_i) - g_{\boldsymbol{x}'}(\boldsymbol{a}_i)| \le (2L_0 + \frac{B_0}{\beta})|\boldsymbol{a}_i^\top \boldsymbol{x} - \boldsymbol{a}_i^\top \boldsymbol{x}'|$$

$$\le (2L_0 + \frac{B_0}{\beta})\|\boldsymbol{a}_i\|_2 \|\boldsymbol{x} - \boldsymbol{x}'\|_2$$

$$= O\big(\sqrt{n}\big[L_0 + \frac{B_0}{\beta}\big]\big)\|\boldsymbol{x} - \boldsymbol{x}'\|_2.$$

- For any $v, v' \in (\mathcal{K}_\epsilon^-)^*$ we have $\left\| |a_i^\top v| - |a_i^\top v'| \right\|_{\psi_2} \leq \|a_i^\top (v - v')\|_{\psi_2} = O(1)\|v - v'\|_2$. Similarly as before, we assume $\|a_i\|_2 = O(\sqrt{n})$, which gives $|a_i^\top v - a_i^\top v'| \leq \|a_i\|_2 \|v - v'\|_2 = O(\sqrt{n})\|v - v'\|_2$. Moreover, $(\mathcal{K}_\epsilon^-)^* \subset \mathbb{B}_2^n$ implies $\left\| |a_i^\top v| \right\|_{\psi_2} = O(1)$ and $|a_i^\top v| \leq \|a_i\|_2 \|v\|_2 = O(\sqrt{n})$ holds for all $v \in (\mathcal{K}_\epsilon^-)^*$.

Combined with Assumption 3, $\mathscr{R}_{u2,c}$ satisfies the conditions of Theorem 2 with

$$M_g \asymp L_0 + \frac{B_0}{\beta}, \ A_g = A_g^{(2)}, \ M_h \asymp 1, \ A_h \asymp O(1)$$

$$L_g \asymp \sqrt{n}\left(L_0 + \frac{B_0}{\beta}\right), \ U_g = U_g^{(2)}, \ L_h \asymp \sqrt{n}, \ U_h \asymp \sqrt{n}$$

and $P_0 = P_0^{(2)} + 2\exp(-\Omega(n))$. Suppose we have

$$m \gtrsim \mathscr{H}\left(\mathcal{K}, \frac{A_g^{(2)}}{\sqrt{mn}[L_0 + \frac{B_0}{\beta}]}\right) + \mathscr{H}\left((\mathcal{K}_\epsilon^-)^*, \frac{A_g^{(2)}}{\sqrt{m}(A_g^{(2)} + \sqrt{n}U_g^{(2)})}\right),$$

which can be guaranteed (from Lemma 6) by

$$m \gtrsim k\log\left(\frac{Lr\sqrt{m}}{A_g^{(2)}}\left[n\left(L_0 + \frac{B_0}{\beta}\right) + \frac{T(A_g^{(2)} + \sqrt{n}U_g^{(2)})}{\epsilon}\right]\right). \tag{C.7}$$

Then, we can invoke Theorem 2 to obtain that the following bound holds with probability at least $1 - mP_0^{(2)} - 2m\exp(-\Omega(n)) - C\exp(-\Omega(k))$:

$$
\begin{aligned}
|\mathscr{R}_{u2,c}| &\lesssim \frac{A_g^{(2)}}{\sqrt{m}}\sqrt{\mathscr{H}\left(\mathcal{K}, \frac{A_g^{(2)}}{\sqrt{mn}[L_0 + \frac{B_0}{\beta}]}\right) + \mathscr{H}\left((\mathcal{K}_\epsilon^-)^*, \frac{A_g^{(2)}}{\sqrt{m}(A_g^{(2)} + \sqrt{n}U_g^{(2)})}\right)} \\
&\lesssim A_g^{(2)}\sqrt{\frac{k}{m}\log\left(\frac{Lr\sqrt{m}}{A_g^{(2)}}\left[n\left(L_0 + \frac{B_0}{\beta}\right) + \frac{T(A_g^{(2)} + \sqrt{n}U_g^{(2)})}{\epsilon}\right]\right)}.
\end{aligned}
\tag{C.8}
$$

### (2.3) Bounding the expectation terms $\mathscr{R}_{u1,e}, \mathscr{R}_{u2,e}$.

Recall that $\xi_{i,\beta}(a) = f_{i,\beta}(a) - Ta$ and $\varepsilon_{i,\beta}(a) = f_{i,\beta}(a) - f_i(a)$, and so $\xi_{i,\beta}(a) = \varepsilon_{i,\beta}(a) + f_i(a) - Ta$. Hence, by using $\mathbb{E}[a_i a_i^\top] = I_n$ and $\|v\|_2 = 1$, we have

$$
\begin{aligned}
\mathscr{R}_{u1,e} &\leq \sup_{x,v} \mathbb{E}\left[\left(f_i(a_i^\top x) - Ta_i^\top x\right)(a_i^\top v)\right] + \sup_{x,v} \mathbb{E}\left[\left[\varepsilon_{i,\beta}(a_i^\top x)\right]a_i^\top v\right] \\
&\leq \sup_{x \in \mathcal{X}} \left\|\mathbb{E}[f_i(a_i^\top x)a_i] - Tx\right\|_2 + \sup_{x,v} \mathbb{E}\left[|\varepsilon_{i,\beta}(a_i^\top x)||a_i^\top v|\right] \\
&\leq \sup_{x \in \mathcal{X}} \rho(x) + \mathscr{R}_{u2,e},
\end{aligned}
\tag{C.9}
$$

where $\rho(x)$ is the model mismatch defined in (2.3), and $\mathscr{R}_{u2,e}$ is defined in (C.3). It remains to bound $\mathscr{R}_{u2,e}$, for which we first apply Cauchy-Schwarz (with $\|v\|_2 \leq 1$) and then use Lemma 13 to obtain

$$
\begin{aligned}
\mathscr{R}_{u2,e} &= \sup_{x,v} \mathbb{E}\left[|\varepsilon_{i,\beta}(a_i^\top x)||a_i^\top v|\right] \\
&\leq \sup_{x,v} \sqrt{\mathbb{E}[|\varepsilon_{i,\beta}(a_i^\top x)|^2]}\sqrt{\mathbb{E}[|a_i^\top v|^2]} \\
&\leq \sup_{x \in \mathcal{K}} \sqrt{\mathbb{E}\left[|\varepsilon_{i,\beta}(a_i^\top x)|^2 \mathbb{1}\left(a_i^\top x \in \mathscr{D}_{f_i} + \left[-\frac{\beta}{2}, \frac{\beta}{2}\right]\right)\right]} \\
&\leq \left(\frac{3L_0\beta}{2} + B_0\right) \sup_{x \in \mathcal{K}} \sqrt{\mathbb{P}\left(a_i^\top x \in \mathscr{D}_{f_i} + \left[-\frac{\beta}{2}, \frac{\beta}{2}\right]\right)} \\
&\leq \left(\frac{3L_0\beta}{2} + B_0\right) \sup_{x \in \mathcal{K}} \sqrt{\mu_\beta(x)},
\end{aligned}
\tag{C.10}
$$

where we use Lemma 13 in the third and fourth line, and $\mu_\beta(\boldsymbol{x})$ is defined in (2.4).

**(3) Combining everything to conclude the proof.**

Recall that in Assumption 4, we assume that

$$\sup_{\boldsymbol{x} \in \mathcal{X}} \rho(\boldsymbol{x}) \lesssim (A_g^{(1)} \vee A_g^{(2)}) \sqrt{\frac{k}{m}},$$

and we take sufficiently small $\beta_1$ such that

$$(L_0 \beta_1 + B_0) \sup_{\boldsymbol{x} \in \mathcal{K}} \sqrt{\mu_{\beta_1}(\boldsymbol{x})} \lesssim (A_g^{(1)} \vee A_g^{(2)}) \sqrt{\frac{k}{m}},$$

then by setting $\beta = \beta_1$, the derived bound of $\mathscr{R}_{u1,c} + \mathscr{R}_{u2,c}$ (see (C.6) and (C.8)) dominates that of $\mathscr{R}_{u1,e} + \mathscr{R}_{u2,e}$ (see (C.9) and (C.10)), and so $\mathscr{R}_{u1} + \mathscr{R}_{u2} \lesssim \mathscr{R}_{u1,c} + \mathscr{R}_{u2,c}$.

Recall that (C.6) and (C.8) are guaranteed by the sample size of (C.5) and (C.7), while (C.5) and (C.7) hold as long as

$$m \gtrsim k \log \left( \frac{Lr\sqrt{m}}{A_g^{(1)} \wedge A_g^{(2)}} \left[ n\left(L_0 + T + \frac{B_0}{\beta_1}\right) + \frac{T(\sqrt{n}(U_g^{(1)} \vee U_g^{(2)}) + (A_g^{(1)} \vee A_g^{(2)}))}{\epsilon} \right] \right)$$

$$:= k\mathscr{L} \qquad \qquad \text{(here we use } \mathscr{L} \text{ to abbreviate the log factors)}$$

(C.11)

with probability at least $1 - m(P_0^{(1)} + P_0^{(2)}) - m \exp(-\Omega(n)) - C \exp(-\Omega(k))$ we have

$$\mathscr{R}_{u1,c} + \mathscr{R}_{u2,c} \lesssim (A_g^{(1)} \vee A_g^{(2)}) \sqrt{\frac{k\mathscr{L}}{m}}.$$

Therefore, the right-hand side of (3.1) can be uniformly bounded by

$$O\left( \|\hat{\boldsymbol{x}} - T\boldsymbol{x}^*\|_2 \cdot (A_g^{(1)} \vee A_g^{(2)}) \sqrt{\frac{k\mathscr{L}}{m}} \right). \tag{C.12}$$

Combining with the uniform lower bound for the left-hand side of (3.1), i.e., $\Omega(\|\hat{\boldsymbol{x}} - T\boldsymbol{x}^*\|_2^2)$, we obtain the following bound uniformly for all $\boldsymbol{x}^*$:

$$\|\hat{\boldsymbol{x}} - T\boldsymbol{x}^*\|_2 \lesssim (A_g^{(1)} \vee A_g^{(2)}) \sqrt{\frac{k\mathscr{L}}{m}}.$$

Hence, as long as

$$m \gtrsim (A_g^{(1)} \vee A_g^{(2)})^2 \frac{k\mathscr{L}}{\epsilon^2},$$

we again obtain $\|\hat{\boldsymbol{x}} - T\boldsymbol{x}^*\|_2 \leq 3\epsilon$, which completes the proof. $\qquad \square$

### C.2 Proof of Theorem 2

*Proof.* **Step 1. Control the process over finite nets.**

Recall that $\mathcal{X}$ and $\mathcal{V}$ are the index sets of $\boldsymbol{x}$ and $\boldsymbol{v}$, as stated in Theorem 2. We first establish the desired concentration for a fixed pair $(\boldsymbol{x}, \boldsymbol{v}) \in \mathcal{X} \times \mathcal{V}$. By Lemma 2, $\|g_{\boldsymbol{x}}(\boldsymbol{a}_i) h_{\boldsymbol{v}}(\boldsymbol{a}_i)\|_{\psi_1} \leq \|g_{\boldsymbol{x}}(\boldsymbol{a}_i)\|_{\psi_2} \|h_{\boldsymbol{v}}(\boldsymbol{a}_i)\|_{\psi_2} \leq A_g A_h$. Furthermore, centering (Lemma 3) gives

$$\|g_{\boldsymbol{x}}(\boldsymbol{a}_i) h_{\boldsymbol{v}}(\boldsymbol{a}_i) - \mathbb{E}[g_{\boldsymbol{x}}(\boldsymbol{a}_i) h_{\boldsymbol{v}}(\boldsymbol{a}_i)]\|_{\psi_1} = O(A_g A_h).$$

Thus, for fixed $(\boldsymbol{x}, \boldsymbol{v}) \in \mathcal{X} \times \mathcal{V}$ we define

$$I_{\boldsymbol{x}, \boldsymbol{v}} = \frac{1}{m} \sum_{i=1}^{m} g_{\boldsymbol{x}}(\boldsymbol{a}_i) h_{\boldsymbol{v}}(\boldsymbol{a}_i) - \mathbb{E}[g_{\boldsymbol{x}}(\boldsymbol{a}_i) h_{\boldsymbol{v}}(\boldsymbol{a}_i)].$$

Then, we can invoke Bernstein's inequality (Lemma 4) to obtain for any $t \geq 0$ that

$$\mathbb{P}(|I_{\boldsymbol{x}, \boldsymbol{v}}| \geq t) \leq 2 \exp\left( -cm \min\left\{ \left(\frac{t}{A_g A_h}\right)^2, \frac{t}{A_g A_h} \right\} \right). \tag{C.13}$$

We construct $\mathcal{G}_1$ as an $\eta_1$-net of $\mathcal{X}$, and $\mathcal{G}_2$ as an $\eta_2$-net of $\mathcal{V}$, with both nets being minimal in that $\log |\mathcal{G}_1| = \mathscr{H}(\mathcal{X}, \eta_1)$, $\log |\mathcal{G}_2| = \mathscr{H}(\mathcal{V}, \eta_2)$, and where $\eta_1, \eta_2$ are to be chosen later. Then, we take a union bound of (C.13) over $(\boldsymbol{x}, \boldsymbol{v}) \in \mathcal{G}_1 \times \mathcal{G}_2$ to obtain

$$\mathbb{P}\left(\sup_{\substack{\boldsymbol{x}\in\mathcal{G}_1 \\ \boldsymbol{v}\in\mathcal{G}_2}} |I_{\boldsymbol{x},\boldsymbol{v}}| \geq t\right) \leq 2\exp\left(\mathscr{H}(\mathcal{X}, \eta_1) + \mathscr{H}(\mathcal{V}, \eta_2) - cm\min\left\{\left(\frac{t}{A_g A_h}\right)^2, \frac{t}{A_g A_h}\right\}\right). \tag{C.14}$$

Now we set $t \asymp A_g A_h \sqrt{\frac{\mathscr{H}(\mathcal{X},\eta_1)+\mathscr{H}(\mathcal{V},\eta_2)}{m}}$ for a sufficiently large hidden constant. Then, if $m \geq C(\mathscr{H}(\mathcal{X}, \eta_1) + \mathscr{H}(\mathcal{V}, \eta_2))$ for large enough $C$ so that $\frac{t}{A_g A_h} \leq 1$ (we assume this now and will confirm it in (C.21) after specifying $\eta_1, \eta_2$), (C.14) gives

$$\mathbb{P}\left(\sup_{\substack{\boldsymbol{x}\in\mathcal{G}_1 \\ \boldsymbol{v}\in\mathcal{G}_2}} |I_{\boldsymbol{x},\boldsymbol{v}}| \gtrsim A_g A_h \sqrt{\frac{\mathscr{H}(\mathcal{X}, \eta_1) + \mathscr{H}(\mathcal{V}, \eta_2)}{m}}\right) \leq 2\exp\left(-\Omega\big(\mathscr{H}(\mathcal{X}, \eta_1) + \mathscr{H}(\mathcal{V}, \eta_2)\big)\right)$$

Hence, from now on we proceed with the proof on the event

$$\sup_{\substack{\boldsymbol{x}\in\mathcal{G}_1 \\ \boldsymbol{v}\in\mathcal{G}_2}} |I_{\boldsymbol{x},\boldsymbol{v}}| \lesssim A_g A_h \sqrt{\frac{\mathscr{H}(\mathcal{X}, \eta_1) + \mathscr{H}(\mathcal{V}, \eta_2)}{m}}, \tag{C.15}$$

which holds within the promised probability.

**Step 2. Control the approximation error of the nets.**

We have derived a bound for $\sup_{\boldsymbol{x}\in\mathcal{G}_1, \boldsymbol{v}\in\mathcal{G}_2} |I_{\boldsymbol{x},\boldsymbol{v}}|$, while we want to control $I = \sup_{\boldsymbol{x}\in\mathcal{X}, \boldsymbol{v}\in\mathcal{V}} |I_{\boldsymbol{x},\boldsymbol{v}}|$, so we further investigate how close these two quantities are. We define the event as

$$\mathscr{E}_1 = \big\{\text{the events in (3.7) hold for all } \boldsymbol{a}_i, i \in [m]\big\},$$

then by assumption **(A2.)** in the theorem statement, a union bound gives $\mathbb{P}(\mathscr{E}_1) \geq 1 - mP_0$. In the following, we proceed with the analysis of the event $\mathscr{E}_1$. Combining with (C.15) we now bound $|I_{\boldsymbol{x},\boldsymbol{v}}|$ for any given $\boldsymbol{x} \in \mathcal{X}, \boldsymbol{v} \in \mathcal{V}$. Specifically, we pick $\boldsymbol{x}' \in \mathcal{G}_1, \boldsymbol{v}' \in \mathcal{G}_2$ such that $\|\boldsymbol{x}' - \boldsymbol{x}\|_2 \leq \eta_1, \|\boldsymbol{v}' - \boldsymbol{v}\|_2 \leq \eta_2$, and thus we have

$$|I_{\boldsymbol{x},\boldsymbol{v}}| \leq |I_{\boldsymbol{x}',\boldsymbol{v}'}| + |I_{\boldsymbol{x},\boldsymbol{v}} - I_{\boldsymbol{x}',\boldsymbol{v}'}| \leq O\left(A_g A_h \sqrt{\frac{\mathscr{H}(\mathcal{X}, \eta_1) + \mathscr{H}(\mathcal{V}, \eta_2)}{m}}\right) + |I_{\boldsymbol{x},\boldsymbol{v}} - I_{\boldsymbol{x}',\boldsymbol{v}'}|. \tag{C.16}$$

Moreover, we have

$$|I_{\boldsymbol{x},\boldsymbol{v}} - I_{\boldsymbol{x}',\boldsymbol{v}'}|$$
$$= \frac{1}{m}\left|\sum_{i=1}^m \Big(g_{\boldsymbol{x}}(\boldsymbol{a}_i)h_{\boldsymbol{v}}(\boldsymbol{a}_i) - g_{\boldsymbol{x}'}(\boldsymbol{a}_i)h_{\boldsymbol{v}'}(\boldsymbol{a}_i)\Big) - m\cdot\mathbb{E}\Big(g_{\boldsymbol{x}}(\boldsymbol{a}_i)h_{\boldsymbol{v}}(\boldsymbol{a}_i) - g_{\boldsymbol{x}'}(\boldsymbol{a}_i)h_{\boldsymbol{v}'}(\boldsymbol{a}_i)\Big)\right|$$
$$\leq \underbrace{\frac{1}{m}\sum_{i=1}^m |g_{\boldsymbol{x}}(\boldsymbol{a}_i)h_{\boldsymbol{v}}(\boldsymbol{a}_i) - g_{\boldsymbol{x}'}(\boldsymbol{a}_i)h_{\boldsymbol{v}'}(\boldsymbol{a}_i)|}_{\text{err}_1} + \underbrace{\mathbb{E}\,|g_{\boldsymbol{x}}(\boldsymbol{a}_i)h_{\boldsymbol{v}}(\boldsymbol{a}_i) - g_{\boldsymbol{x}'}(\boldsymbol{a}_i)h_{\boldsymbol{v}'}(\boldsymbol{a}_i)|}_{\text{err}_2}$$
$$\tag{C.17}$$

We bound $\text{err}_1$ using the event $\mathscr{E}_1$ as follows:

$$\text{err}_1 \leq \frac{1}{m}\sum_{i=1}^m \Big||g_{\boldsymbol{x}}(\boldsymbol{a}_i) - g_{\boldsymbol{x}'}(\boldsymbol{a}_i)|\cdot|h_{\boldsymbol{v}}(\boldsymbol{a}_i)| + |h_{\boldsymbol{v}}(\boldsymbol{a}_i) - h_{\boldsymbol{v}'}(\boldsymbol{a}_i)|\cdot|g_{\boldsymbol{x}'}(\boldsymbol{a}_i)|\Big|$$
$$\leq \frac{1}{m}\sum_{i=1}^m \Big|L_g\cdot\|\boldsymbol{x} - \boldsymbol{x}'\|_2\cdot U_h + L_h\cdot\|\boldsymbol{v} - \boldsymbol{v}'\|_2\cdot U_g\Big| \tag{C.18}$$
$$\leq \frac{1}{m}\sum_{i=1}^m \Big|L_g U_h \eta_1 + L_h U_g \eta_2\Big| = L_g U_h \eta_1 + L_h U_g \eta_2.$$

On the other hand, we bound $\mathrm{err}_2$ using assumption **(A1.)**in the theorem statement. Noting that $\mathbb{E}|X| = O(\|X\|_{\psi_1})$ [50, Proposition 2.7.1(b)], and further applying Lemma 2 we obtain

$$
\begin{aligned}
\mathrm{err}_2 &\lesssim \|g_{\boldsymbol{x}}(\boldsymbol{a}_i)h_{\boldsymbol{v}}(\boldsymbol{a}_i) - g_{\boldsymbol{x}'}(\boldsymbol{a}_i)h_{\boldsymbol{v}'}(\boldsymbol{a}_i)\|_{\psi_1}\\
&\leq \|(g_{\boldsymbol{x}}(\boldsymbol{a}_i) - g_{\boldsymbol{x}'}(\boldsymbol{a}_i))h_{\boldsymbol{v}}(\boldsymbol{a}_i)\|_{\psi_1} + \|g_{\boldsymbol{x}'}(\boldsymbol{a}_i)(h_{\boldsymbol{v}}(\boldsymbol{a}_i) - h_{\boldsymbol{v}'}(\boldsymbol{a}_i))\|_{\psi_1}\\
&\leq \|g_{\boldsymbol{x}}(\boldsymbol{a}_i) - g_{\boldsymbol{x}'}(\boldsymbol{a}_i)\|_{\psi_2}\|h_{\boldsymbol{v}}(\boldsymbol{a}_i)\|_{\psi_2} + \|g_{\boldsymbol{x}'}(\boldsymbol{a}_i)\|_{\psi_2}\|h_{\boldsymbol{v}}(\boldsymbol{a}_i) - h_{\boldsymbol{v}'}(\boldsymbol{a}_i)\|_{\psi_2}\\
&\leq M_g \cdot \|\boldsymbol{x} - \boldsymbol{x}'\|_2 \cdot A_h + A_g \cdot M_h \cdot \|\boldsymbol{v} - \boldsymbol{v}'\|_2 \leq M_g A_h \eta_1 + A_g M_h \eta_2.
\end{aligned}
\tag{C.19}
$$

Note that the bounds (C.18) and (C.19) hold uniformly for all $(\boldsymbol{x}, \boldsymbol{v}) \in \mathcal{X} \times \mathcal{V}$, and hence, we can substitute them into (C.16) and (C.17) to obtain

$$
\sup_{\substack{\boldsymbol{x}\in\mathcal{X}\\ \boldsymbol{v}\in\mathcal{V}}} |I_{\boldsymbol{x},\boldsymbol{v}}| \leq O\left(A_g A_h \sqrt{\frac{\mathscr{H}(\mathcal{X},\eta_1) + \mathscr{H}(\mathcal{V},\eta_2)}{m}}\right) + (L_g U_h + M_g A_h)\eta_1 + (L_h U_g + M_h A_g)\eta_2.
\tag{C.20}
$$

Recall that we use the shorthand $S_{g,h} := L_g U_h + M_g A_h$ and $T_{g,h} := L_h U_g + M_h A_g$. We set $\eta_1 \asymp \frac{A_g A_h}{\sqrt{m}S_{g,h}}$, $\eta_2 \asymp \frac{A_g A_h}{\sqrt{m}T_{g,h}}$ so that the right-hand side of (C.20) is dominated by the first term. Overall, with a sample size satisfying

$$
m = \Omega\left(\mathscr{H}\left(\mathcal{X}, \frac{A_g A_h}{\sqrt{m}S_{g,h}}\right) + \mathscr{H}\left(\mathcal{V}, \frac{A_g A_h}{\sqrt{m}T_{g,h}}\right)\right),
\tag{C.21}
$$

we can bound $I = \sup_{\boldsymbol{x}\in\mathcal{X}} \sup_{\boldsymbol{v}\in\mathcal{V}} |I_{\boldsymbol{x},\boldsymbol{v}}|$ (defined in the theorem statement) as

$$
I \lesssim A_g A_h \sqrt{\frac{\mathscr{H}(\mathcal{X}, A_g A_h m^{-1/2} S_{g,h}^{-1}) + \mathscr{H}(\mathcal{V}, A_g A_h m^{-1/2} T_{g,h}^{-1})}{m}}
\tag{C.22}
$$

with probability at least

$$
1 - mP_0 - 2\exp\left[-\Omega\left(\mathscr{H}\left(\mathcal{X}, \frac{A_g A_h}{\sqrt{m}S_{g,h}}\right) + \mathscr{H}\left(\mathcal{V}, \frac{A_g A_h}{\sqrt{m}T_{g,h}}\right)\right)\right].
$$

This completes the proof. $\qquad\square$

## D  Other Omitted Proofs

### D.1  Proof of Lemma 1 (Lipschitz continuity of $f_{i,\beta}$ and $\varepsilon_{i,\beta}$).

*Proof.* It is straightforward to check that $f_{i,\beta}$ and $|\varepsilon_{i,\beta}|$ are piece-wise continuous functions; hence, it suffices to prove that they are Lipschitz with the claimed Lipschitz constant over each piece. In any interval contained in the part of $x \notin \mathscr{D}_{f_i} + [-\frac{\beta}{2}, \frac{\beta}{2}]$, $f_{i,\beta} = f_i$ and $|\varepsilon_{i,\beta}| = 0$ trivially satisfy the claim. In any interval contained in $[x_0 - \frac{\beta}{2}, x_0]$ for some $x_0 \in \mathscr{D}_{f_i}$, $f_{i,\beta}$ is linear with slope $\frac{2}{\beta}\left(f_i^a(x_0) - f_i(x_0 - \frac{\beta}{2})\right)$, combined with the bound $|f_i^a(x_0) - f_i(x_0 - \frac{\beta}{2})| \leq |f_i^a(x_0) - f_i^-(x_0)| + |f_i^-(x_0) - f_i(x_0 - \frac{\beta}{2})| \leq \frac{|f_i^+(x_0) - f_i^-(x_0)|}{2} + \frac{L_0 \beta}{2} \leq \frac{1}{2}\left(B_0 + L_0\beta\right)$, we know that $f_{i,\beta}$ is $\left(L_0 + \frac{B_0}{\beta}\right)$-Lipschitz. Further, $|\varepsilon_{i,\beta}| = |f_{i,\beta} - f_i|$, and $f_i$ is $L_0$-Lipschitz over this interval, so $|\varepsilon_{i,\beta}|$ is $\left(2L_0 + \frac{B_0}{\beta}\right)$-Lipschitz continuous. A similar argument applies to an interval contained in $[x_0, x_0 + \frac{\beta}{2}]$. $\qquad\square$

### D.2  Proof of Lemma 6 (Metric entropy of constraint sets).

*Proof.* **Bounding $\mathscr{H}(\mathcal{K}, \eta)$.**

By [50, Corollary 4.2.13], there exists an $\left(\frac{\eta}{Lr}\right)$-net $\mathcal{G}_1$ of $\mathbb{B}_2^k$ such that

$$
\log|\mathcal{G}_1| \leq k\log\left(\frac{2Lr}{\eta} + 1\right) \leq k\log\frac{3Lr}{\eta},
$$

where we use $\eta \leq Lr$. Note that $r\mathcal{G}_1$ is an $\left(\frac{\eta}{L}\right)$-net of $\mathbb{B}_2^k(r)$, and because $G$ is $L$-Lipschitz, $G(r\mathcal{G}_1)$ is an $\eta$-net of $\mathcal{K}$, thus yielding $\mathscr{H}(\mathcal{K}, \eta) \leq k\log\frac{3Lr}{\eta}$.

**Bounding $\mathscr{H}(\mathcal{K}^-, \eta)$ and $\mathscr{H}(\mathcal{K}_\epsilon^-, \eta)$.**

We construct $\mathcal{G}_2$ as an $\left(\frac{\eta}{2}\right)$-net of $\mathcal{K}$ satisfying $\log|\mathcal{G}_2| \le k\log\frac{6Lr}{\eta}$. Then, it is easy to see that $\mathcal{G}_2 - \mathcal{G}_2$ is an $\eta$-net of $\mathcal{K}^- = \mathcal{K} - \mathcal{K}$, showing that

$$\mathscr{H}(\mathcal{K}^-, \eta) \le \log|\mathcal{G}_2|^2 \le 2k\log\frac{6Lr}{\eta}.$$

For a given $T > 0$, this directly implies $\mathscr{H}(T\mathcal{K}^-, \eta) \le 2k\log\frac{6TLr}{\eta}$. Moreover, because $\mathcal{K}_\epsilon^- \subset T\mathcal{K}^-$, by [50, Exercise 4.2.10] (which states that $\mathscr{H}(\mathcal{K}_1, r) \le \mathscr{H}(\mathcal{K}_2, \frac{r}{2})$ holds for any $r > 0$ if $\mathcal{K}_1 \subset \mathcal{K}_2$) we obtain

$$\mathscr{H}(\mathcal{K}_\epsilon^-, \eta) \le \mathscr{H}\left(T\mathcal{K}^-, \frac{\eta}{2}\right) \le 2k\log\frac{12TLr}{\eta}.$$

**Bounding $\mathscr{H}\big((\mathcal{K}_\epsilon^-)^*, \eta\big)$.**

We construct $\mathcal{G}_3$ as an $(\epsilon\eta)$-net of $\mathcal{K}_\epsilon^-$ satisfying $\log|\mathcal{G}_3| \le 2k\log\frac{12TLr}{\epsilon\eta}$, then we consider $(\mathcal{G}_3)^* := \left\{\frac{z}{\|z\|_2} : z \in \mathcal{G}_3\right\}$. We aim to prove that $(\mathcal{G}_3)^*$ is an $\eta$-net of $(\mathcal{K}_\epsilon^-)^*$. Note that any $x_1 \in (\mathcal{K}_\epsilon^-)^*$ can be written as $\frac{z_1}{\|z_1\|_2}$ for some $z_1 \in \mathcal{K}_\epsilon^-$ and recall that $\|z_1\|_2 \ge 2\epsilon$. Moreover, by construction, there exists some $z_2 \in \mathcal{G}_3$ such that $\|z_1 - z_2\|_2 \le \epsilon\eta$. Note that $\frac{z_2}{\|z_2\|_2} \in (\mathcal{G}_3)^*$, and moreover we have

$$\left\|\frac{z_1}{\|z_1\|_2} - \frac{z_2}{\|z_2\|_2}\right\|_2 \le \left\|\frac{z_1}{\|z_1\|_2} - \frac{z_2}{\|z_1\|_2}\right\|_2 + \left\|\frac{z_2}{\|z_1\|_2} - \frac{z_2}{\|z_2\|_2}\right\|_2$$
$$= \frac{\|z_1 - z_2\|_2}{\|z_1\|_2} + \frac{|\|z_2\|_2 - \|z_1\|_2|}{\|z_1\|_2}$$
$$\le \frac{2\|z_1 - z_2\|_2}{\|z_1\|_2} \le \frac{2\epsilon\eta}{2\epsilon} = \eta.$$

Hence, we obtain

$$\mathscr{H}\big((\mathcal{K}_\epsilon^-)^*, \eta\big) \le \log|(\mathcal{G}_3)^*| \le \log|\mathcal{G}_3| \le 2k\log\frac{12TLr}{\epsilon\eta},$$

which completes the proof. $\qquad\square$

### D.3 Proof of Lemma 8 (Choice of $T$ in 1-bit GCS).

*Proof.* Since $x \in \mathbb{S}^{n-1}$, for some orthogonal matrix $P$ we have $Px = e_1$ (the first column of $I_n$). Since $\tilde{a} := Pa = [\tilde{a}_i]$ has the same distribution as $a$, we have

$$\mathbb{E}[\text{sign}(a^\top x)a] = \mathbb{E}[\text{sign}(\tilde{a}^\top e_1)P^\top\tilde{a}] = P^\top\mathbb{E}[\text{sign}(\tilde{a}_1)\tilde{a}]$$
$$= P^\top\sqrt{\frac{2}{\pi}}e_1 = \sqrt{\frac{2}{\pi}}x.$$

$\qquad\square$

### D.4 Proof of Lemma 9 (Choice of $T$ in 1-bit GCS with dithering).

*Proof.* We first note that

$$\left\|\mathbb{E}[\text{sign}(a^\top x + \tau)a] - \frac{x}{\lambda}\right\|_2 = \frac{1}{\lambda}\sup_{v \in \mathbb{S}^{n-1}}\left(\mathbb{E}[\lambda \cdot \text{sign}(a^\top x + \tau)a^\top v] - x^\top v\right). \tag{D.1}$$

We first fix $a$ and expect over $\tau \sim \mathscr{U}[-\lambda, \lambda]$ to obtain

$$\mathbb{E}_\tau\left[\lambda\,\text{sign}(a^\top x + \tau)a^\top v\right]$$
$$= (\lambda a^\top v)\left(\mathbb{1}(|a^\top x| > \lambda)\,\text{sign}(a^\top x) + \mathbb{1}(|a^\top x| \le \lambda) \cdot \left(\frac{\lambda + a^\top x}{2\lambda} - \frac{\lambda - a^\top x}{2\lambda}\right)\right)$$
$$= (a^\top x)(a^\top v)\mathbb{1}(|a^\top x| \le \lambda) + (\lambda a^\top v)\,\text{sign}(a^\top x)\mathbb{1}(|a^\top x| > \lambda).$$

We plug this into (D.1), and note that $\boldsymbol{x}^\top \boldsymbol{v} = \mathbb{E}[(\boldsymbol{a}^\top \boldsymbol{x})(\boldsymbol{a}^\top \boldsymbol{v})]$, which gives

$$
\left\| \mathbb{E}[\mathrm{sign}(\boldsymbol{a}^\top \boldsymbol{x} + \tau)\boldsymbol{a}] - \frac{\boldsymbol{x}}{\lambda} \right\|_2
$$
$$
= \frac{1}{\lambda} \sup_{\boldsymbol{v} \in \mathbb{S}^{n-1}} \mathbb{E}\left( \left[ (\lambda \boldsymbol{a}^\top \boldsymbol{v})\,\mathrm{sign}(\boldsymbol{a}^\top \boldsymbol{x}) - (\boldsymbol{a}^\top \boldsymbol{x})(\boldsymbol{a}^\top \boldsymbol{v}) \right] \mathbb{1}(|\boldsymbol{a}^\top \boldsymbol{x}| > \lambda) \right) \tag{D.2}
$$
$$
\leq \frac{1}{\lambda} \sup_{\boldsymbol{v} \in \mathbb{S}^{n-1}} \mathbb{E}\left( [\lambda|\boldsymbol{a}^\top \boldsymbol{v}| + |\boldsymbol{a}^\top \boldsymbol{x}||\boldsymbol{a}^\top \boldsymbol{v}|]\mathbb{1}(|\boldsymbol{a}^\top \boldsymbol{x}| > \lambda) \right)
$$

For any $\boldsymbol{x} \in \mathbb{B}_2^n(R)$ and $\boldsymbol{v} \in \mathbb{S}^{n-1}$, we have $\|\boldsymbol{a}^\top \boldsymbol{x}\|_{\psi_2} = O(R)$ and $\|\boldsymbol{a}^\top \boldsymbol{v}\|_{\psi_2} = O(1)$. Applying the Cauchy-Schwarz inequality, we obtain

$$
\mathbb{E}\left([\lambda|\boldsymbol{a}^\top \boldsymbol{v}| + |\boldsymbol{a}^\top \boldsymbol{x}||\boldsymbol{a}^\top \boldsymbol{v}|]\mathbb{1}(|\boldsymbol{a}^\top \boldsymbol{x}| > \lambda)\right)
$$
$$
\leq \sqrt{\mathbb{E}[(\lambda|\boldsymbol{a}^\top \boldsymbol{v}| + |\boldsymbol{x}^\top \boldsymbol{a}\boldsymbol{a}^\top \boldsymbol{v}|)^2]}\sqrt{\mathbb{P}(|\boldsymbol{a}^\top \boldsymbol{x}| > \lambda)}
$$
$$
\leq \sqrt{2\big(\lambda^2 \mathbb{E}[|\boldsymbol{a}^\top \boldsymbol{v}|^2] + \mathbb{E}[(\boldsymbol{a}^\top \boldsymbol{x})^2(\boldsymbol{a}^\top \boldsymbol{v})^2]\big)}\sqrt{2\exp(-c\lambda^2/R^2)}
$$
$$
\lesssim \lambda \exp\left(-\frac{c\lambda^2}{R^2}\right) \lesssim \frac{\lambda}{m^9}
$$

Note that in the third line, we use the probability tail bound of the sub-Gaussian $|\boldsymbol{a}^\top \boldsymbol{x}|$, and in the last line, we use $\lambda = CR\sqrt{\log m}$ with some sufficiently large $C$. The proof is completed by substituting this into (D.2). $\qquad \square$

### D.5  Proof of Lemma 10 (Choice of $T$ in SIM).

*Proof.* This lemma slightly generalizes that of Lemma 8. We again choose an orthogonal matrix $\boldsymbol{P}$ such that $\boldsymbol{P}\boldsymbol{x} = \boldsymbol{e}_1$, where $\boldsymbol{e}_1$ represents the first column of $\boldsymbol{I}_n$. Since $\boldsymbol{a}$ and $\boldsymbol{P}\boldsymbol{a}$ have the same distribution, we have

$$
\mathbb{E}[f(\boldsymbol{a}^\top \boldsymbol{x})\boldsymbol{a}] = \boldsymbol{P}^\top \mathbb{E}[f((\boldsymbol{P}\boldsymbol{a})^\top \boldsymbol{e}_1)\boldsymbol{P}\boldsymbol{a}]
$$
$$
= \boldsymbol{P}^\top \mathbb{E}[f(\boldsymbol{a}^\top \boldsymbol{e}_1)\boldsymbol{a}] = \boldsymbol{P}^\top (\mu \boldsymbol{e}_1) = \mu \boldsymbol{x}.
$$

$\qquad \square$

### D.6  Proof of Lemma 11 (Choice of $T$ in uniformly quantized GCS with dithering).

*Proof.* In the theorem, the statement before "In particular" can be found in [18, Theorem 1]. Based on this, we have $\mathbb{E}[\mathcal{Q}_\delta(\boldsymbol{a}^\top \boldsymbol{x} + \tau)\boldsymbol{a}] = \mathbb{E}_{\boldsymbol{a}}\mathbb{E}_\tau[\mathcal{Q}_\delta(\boldsymbol{a}^\top \boldsymbol{x} + \tau)\boldsymbol{a}] = \mathbb{E}_{\boldsymbol{a}}(\boldsymbol{a}\boldsymbol{a}^\top \boldsymbol{x}) = \boldsymbol{x}$. $\qquad \square$

### D.7  Proof of Lemma 12. (Bounds on $|\xi_{i,\beta}|$ and $|\varepsilon_{i,\beta}|$ for the uniform quantizer).

*Proof.* By the definition of $f_{i,\beta}$ in (3.4), we have $|\varepsilon_{i,\beta}(a)| = |f_{i,\beta}(a) - f_i(a)| \leq \delta$. It follows that $f_i(\cdot) = \mathcal{Q}_\delta(\cdot + \tau)$ with $\mathcal{Q}_\delta(a) = \delta\left(\lfloor \frac{a}{\delta} \rfloor + \frac{1}{2}\right)$, and $|\mathcal{Q}_\delta(a) - a| \leq \frac{\delta}{2}$ holds for any $a \in \mathbb{R}$. Hence, we have $|f_i(a) - a| = |\mathcal{Q}_\delta(a+\tau) - (a+\tau) + \tau| \leq |\mathcal{Q}_\delta(a+\tau) - (a+\tau)| + |\tau| \leq \frac{\delta}{2} + \frac{\delta}{2} = \delta$. To complete the proof, we use the inequalities $|\xi_{i,\beta}(a)| \leq |f_{i,\beta}(a) - f_i(a)| + |f_i(a) - a| \leq \delta + \delta = 2\delta$. $\qquad \square$

### D.8  Proof of Lemma 13. (Bound on the approximation error $|\varepsilon_{i,\beta}|$)

*Proof.* For any $a \notin \mathscr{D}_{f_i} + [-\frac{\beta}{2}, \frac{\beta}{2}]$, by the definition in (3.4) we have $\varepsilon_{i,\beta}(a) = 0$. If $a \in [x_0 - \frac{\beta}{2}, x_0]$ for some $x_0 \in \mathscr{D}_{f_i}$, then we have

$$
|\varepsilon_{i,\beta}(a)| = |f_{i,\beta}(a) - f_i(a)|
$$
$$
\leq |f_{i,\beta}(a) - f_{i,\beta}(x_0)| + |f_{i,\beta}(x_0) - f_i^-(x_0)| + |f_i^-(x_0) - f_i(a)|
$$
$$
\leq \left(2L_0 + \frac{B_0}{\beta}\right)|a - x_0| + |f_i^a(x_0) - f_i^-(x_0)| + L_0|x_0 - a|
$$
$$
\leq \left(3L_0 + \frac{B_0}{\beta}\right) \cdot \frac{\beta}{2} + \frac{1}{2}|f_i^+(x_0) - f_i^-(x_0)| \leq \frac{3L_0\beta}{2} + B_0,
$$

where we use Lemma 1 and Assumption 2 in the third line, and use $|a - x_0| \leq \frac{\beta}{2}$ and $f_i^a(x_0) = \frac{1}{2}\left(f_i^-(x_0) + f_i^+(x_0)\right)$ in the fourth line. $\qquad \square$

# E  Parameter Selection for Specific Models

## E.1  1-bit GCS

To specialize Theorem 1 to this model, we select the parameters as follows:

- **Assumption 2.** Under the 1-bit observation model $y_i = \text{sign}(\boldsymbol{a}_i^\top \boldsymbol{x}^*)$, the function $f_i(\cdot) = f(\cdot) = \text{sign}(\cdot)$ satisfies Assumption 2 with $(B_0, L_0, \beta_0) = (2, 0, \infty)$.
- **(2.5) in Assumption 4.** Recall that $\mathcal{K} \subset \mathbb{S}^{n-1}$. Under the assumption $\|\boldsymbol{x}^*\|_2 = 1$, we set $T = \sqrt{2/\pi}$ so that $\rho(\boldsymbol{x}) = 0$ holds for all $\boldsymbol{x} \in \mathcal{X}$ (Lemma 8), which provides (2.5).
- **Assumption 3.** By Lemma 7, we have $\mathbb{P}(\|\boldsymbol{a}\|_2 = O(\sqrt{n})) \geq 1 - 2\exp(-\Omega(n))$, and we suppose that this high-probability event holds. Also note that $|f_{i,\beta}| \leq 1$. Hence, we have

$$\|\xi_{i,\beta}(\boldsymbol{a}^\top \boldsymbol{x})\|_{\psi_2} \leq \|f_{i,\beta}(\boldsymbol{a}^\top \boldsymbol{x})\|_{\psi_2} + \|T\boldsymbol{a}^\top \boldsymbol{x}\|_{\psi_2} = O(1),$$
$$\sup_{\boldsymbol{x} \in \mathcal{K}} |\xi_{i,\beta}(\boldsymbol{a}^\top \boldsymbol{x})| \leq |f_{i,\beta}(\boldsymbol{a}^\top \boldsymbol{x})| + |T\boldsymbol{a}^\top \boldsymbol{x}| \leq 1 + T\|\boldsymbol{a}\|_2 = O(\sqrt{n}).$$

Because $\varepsilon_{i,\beta} = f_{i,\beta} - f_i$, we have $\|\varepsilon_{i,\beta}(\boldsymbol{a}^\top \boldsymbol{x})\|_{\psi_2} = O(1)$, and $|\varepsilon_{i,\beta}(\boldsymbol{a}^\top \boldsymbol{x})| \leq 2$ holds deterministically. Hence, regarding the parameters in Assumption 3, we can take

$$A_g^{(1)} \asymp 1, U_g^{(1)} \asymp \sqrt{n}, P_0^{(1)} \asymp \exp(-\Omega(n)), A_g^{(2)} \asymp 1, U_g^{(2)} \asymp 1, P_0^{(2)} = 0.$$

- **(2.6) in Assumption 4.** It remains to pick $\beta_1$ that satisfies (2.6). Note that $\mathscr{D}_{f_i} = \{0\}$, and for any $\boldsymbol{x} \in \mathcal{K}$, $\boldsymbol{a}^\top \boldsymbol{x} \sim \mathcal{N}(0, 1)$, so we have

$$\mu_\beta(\boldsymbol{x}) = \mathbb{P}\left(\boldsymbol{a}^\top \boldsymbol{x} \in \left[-\frac{\beta}{2}, \frac{\beta}{2}\right]\right) = O(\beta).$$

Thus, we take $\beta = \beta_1 \asymp \frac{k}{m}$ to guarantee (2.6).

## E.2  1-bit GCS with dithering

To specialize Theorem 1 to this model, we select the parameters as follows:

- **Assumption 2.** The observation function can be written as $f(\cdot) = \text{sign}(\cdot + \tau)$ with $\tau \sim \mathscr{U}[-\lambda, \lambda]$, which satisfies Assumption 2 with $(B_0, L_0, \beta_0) = (2, 0, \infty)$.
- **(2.5) in Assumption 4.** We set $\lambda = CR\sqrt{\log m}$ with $C$ large enough, so that Lemma 9 justifies (2.5).
- **Assumption 3.** By Lemma 7, we have $\mathbb{P}(\|\boldsymbol{a}\|_2 = O(\sqrt{n})) \geq 1 - 2\exp(-\Omega(n))$. Assume this event holds, and note that $f_{i,\beta}$ is still bounded by 1, we have

$$\|\xi_{i,\beta}(\boldsymbol{a}^\top \boldsymbol{x})\|_{\psi_2} \leq \|f_{i,\beta}(\boldsymbol{a}^\top \boldsymbol{x})\|_{\psi_2} + \|\lambda^{-1}\boldsymbol{a}^\top \boldsymbol{x}\|_{\psi_2} = O(R/\lambda) + O(1) = O(1),$$
$$\sup_{\boldsymbol{x} \in \mathcal{K}} |\xi_{i,\beta}(\boldsymbol{a}^\top \boldsymbol{x})| \leq 1 + \sup_{\boldsymbol{x} \in \mathcal{K}} |\lambda^{-1}\boldsymbol{a}^\top \boldsymbol{x}| \leq 1 + \sup_{\boldsymbol{x} \in \mathcal{K}} \lambda^{-1}\|\boldsymbol{a}\|_2 \|\boldsymbol{x}\|_2 = O(\sqrt{n}).$$

Moreover, because $\varepsilon_{i,\beta} = f_{i,\beta} - f_i$, the following hold deterministically: $\|\varepsilon_{i,\beta}(\boldsymbol{a}^\top \boldsymbol{x})\|_{\psi_2} = O(1)$, $\sup_{\boldsymbol{x} \in \mathcal{K}} |\varepsilon_{i,\beta}(\boldsymbol{a}^\top \boldsymbol{x})| = O(1)$. Thus, regarding the parameters in Assumption 3 we can take

$$A_g^{(1)} \asymp 1, \ U_g^{(1)} \asymp \sqrt{n}, \ P_0^{(1)} \asymp \exp(-\Omega(n)), \ A_g^{(2)} \asymp 1, \ U_g^{(2)} \asymp 1, \ P_0^{(2)} = 0.$$

- **(2.6) in Assumption 4.** It remains to confirm (2.6) for suitable $\beta_1$. For any $\beta$, note that $\mathscr{D}_{f_i} + [-\frac{\beta}{2}, \frac{\beta}{2}] = [\tau - \frac{\beta}{2}, \tau + \frac{\beta}{2}]$, and hence for any $\boldsymbol{x} \in \mathcal{K} \subset \mathbb{B}_2^n(R)$ we have

$$\mu_\beta(\boldsymbol{x}) = \mathbb{P}\left(\boldsymbol{a}^\top \boldsymbol{x} \in \left[-\tau - \frac{\beta}{2}, -\tau + \frac{\beta}{2}\right]\right) = \mathbb{P}\left(\boldsymbol{a}^\top \boldsymbol{x} + \tau \in \left[-\frac{\beta}{2}, \frac{\beta}{2}\right]\right) \leq \frac{\beta}{\lambda},$$

which can be seen by conditioning on $\boldsymbol{a}$. Hence, we can take $\beta = \beta_1 = \frac{\lambda k}{m}$ to guarantee (2.6).

### E.3 Lipschitz-continuous SIM with generative prior

To specialize Theorem 1 to this model, we select the parameters as follows:

- **Assumption 2.** Since $f$ is $\hat{L}$-Lipschitz by assumption, it satisfies Assumption 2 with $(B_0, L_0, \beta_0) = (0, \hat{L}, \infty)$.
- **(2.5) in Assumption 4.** Recall that we have defined the quantities $\mu = \mathbb{E}[f(g)g], \psi = \|f(g)\|_{\psi_2}$, where $g \sim \mathcal{N}(0, 1)$. Then, we choose $T = \mu$ so that $\rho(\boldsymbol{x}) = 0$ holds for any $\boldsymbol{x}$ (Lemma 10), thus justifying (2.5).
- **Assumption 3.** Because $f_i$ is $\hat{L}$-Lipschitz and does not contain any discontinuity, there is no need to construct the Lipschitz approximation $f_{i,\beta}$ for some $\beta > 0$, while we simply use $\beta = 0$, which implies $f_{i,\beta} = f_i$ and $\varepsilon_{i,\beta} = 0$. Note that $\xi_{i,\beta}(a) = f_i(a) - \mu a$, and so we have

$$\|f_i(\boldsymbol{a}^\top \boldsymbol{x}) - \mu \boldsymbol{a}^\top \boldsymbol{x}\|_{\psi_2} \leq \|f_i(\boldsymbol{a}^\top \boldsymbol{x})\|_{\psi_2} + \|\mu \boldsymbol{a}^\top \boldsymbol{x}\|_{\psi_2} = O(\psi + \mu).$$

We suppose $\|\boldsymbol{a}\|_2 = O(\sqrt{n})$, which holds with probability at least $1 - 2\exp(-\Omega(n))$ (Lemma 7); we also suppose $f_i(0) \leq \hat{B}$, which holds with probability at least $1 - P_0'$ by assumption. On these two events, we have

$$|f_i(\boldsymbol{a}^\top \boldsymbol{x}) - \mu \boldsymbol{a}^\top \boldsymbol{x}| \leq |f_i(\boldsymbol{a}^\top \boldsymbol{x}) - f_i(0)| + |f_i(0)| + \mu\|\boldsymbol{a}\|_2\|\boldsymbol{x}\|_2$$
$$\leq \hat{L}\|\boldsymbol{a}\|_2 + \hat{B} + \mu\|\boldsymbol{a}\|_2 \lesssim (\hat{L} + \mu)\sqrt{n} + \hat{B}.$$

Combined with $\varepsilon_{i,\beta} = 0$, we can set the parameters in Assumption 3 as follows:

$$A_g^{(1)} \asymp \psi + \mu, \ U_g^{(1)} \asymp (\hat{L} + \mu)\sqrt{n} + \hat{B}, \ P_0^{(1)} \asymp P_0' + \exp(-\Omega(n)),$$
$$A_g^{(2)} \asymp \psi + \mu, \ U_g^{(2)} = 0, \ P_0^{(2)} = 0.$$

- **(2.6) in Assumption 4.** Because $\beta = 0$ and $\mathscr{D}_{f_i} = \varnothing$, (2.6) is trivially satisfied.

### E.4 Uniformly quantized GCS with dithering

To specialize Theorem 1 to this model, we select the parameters as follows:

- **Assumption 2.** The uniform quantizer with resolution $\delta > 0$ is defined as $\mathcal{Q}_\delta(a) = \delta\left(\lfloor \frac{a}{\delta} \rfloor + \frac{1}{2}\right)$ for $a \in \mathbb{R}$. We consider this quantizer with dithering $\tau_i \sim \mathscr{U}[-\frac{\delta}{2}, \frac{\delta}{2}]$. Specifically, we observe $y_i = \mathcal{Q}_\delta(\boldsymbol{a}_i^\top \boldsymbol{x}^* + \tau_i)$, so the observation function is $f(\cdot) = \mathcal{Q}_\delta(\cdot + \tau)$ with $\tau \sim \mathscr{U}[-\frac{\delta}{2}, \frac{\delta}{2}]$. Hence, Assumption 2 is satisfied with $(B_0, L_0, \beta_0) = (\delta, 0, \delta)$.
- **(2.5) in Assumption 4.** The benefit of dithering is to whiten the quantization noise. With $T = 1$, for any $\boldsymbol{x} \in \mathcal{K}$, Lemma 11 implies $\rho(\boldsymbol{x}) = \|\mathbb{E}[\mathcal{Q}_\delta(\boldsymbol{a}_i^\top \boldsymbol{x} + \tau_i)\boldsymbol{a}_i] - \boldsymbol{x}\|_2 = 0$, thus justifying (2.5).
- **Assumption 3.** Note that for any $\beta \in (0, \frac{\delta}{2})$, by Lemma 12, we can take the parameters for Assumption 3 as follows:

$$A_g^{(1)}, U_g^{(1)}, A_g^{(2)}, U_g^{(2)} \asymp \delta, \ P_0^{(1)} = P_0^{(2)} = 0.$$

- **(2.6) in Assumption 4.** All that remains is to pick $\beta = \beta_1$ that satisfies (2.6). Because $\mathscr{D}_{f_i} = -\tau_i + \delta\mathbb{Z}$, hence for any $\boldsymbol{x} \in \mathcal{K}$ we have

$$\mu_\beta(\boldsymbol{x}) = \mathbb{P}\left(\boldsymbol{a}^\top \boldsymbol{x} \in -\tau + \delta\mathbb{Z} + \left[-\frac{\beta}{2}, \frac{\beta}{2}\right]\right) = \mathbb{P}\left(\boldsymbol{a}^\top \boldsymbol{x} + \tau \in \delta\mathbb{Z} + \left[-\frac{\beta}{2}, \frac{\beta}{2}\right]\right) = O\left(\frac{\beta}{\delta}\right),$$

which can be seen by using the randomness of $\tau \sim \mathscr{U}[-\frac{\delta}{2}, \frac{\delta}{2}]$ conditionally on $\boldsymbol{x}$. Hence, we take $\beta = \beta_1 \asymp \frac{k\delta}{m}$, which provides (2.6).

## F Handling Sub-Gaussian Additive Noise

In this appendix, we describe how our results can be extended to the noisy model $\boldsymbol{y} = \boldsymbol{f}(\boldsymbol{A}\boldsymbol{x}^*) + \boldsymbol{\eta}$, where $\boldsymbol{\eta} \in \mathbb{R}^m$ is the noise vector that is independent of $(\boldsymbol{A}, \boldsymbol{f})$ and has i.i.d. sub-Gaussian entries $\eta_i$ satisfying $\|\eta_i\|_{\psi_2} = O(1)$. Along similar lines as in (3.1)-(3.3), we find that $\boldsymbol{\eta}$ gives rise to an additional term $\frac{2}{m}\langle \boldsymbol{\eta}, \boldsymbol{A}(\hat{\boldsymbol{x}} - T\boldsymbol{x}^*)\rangle$ to the right-hand side of (3.1), which is bounded by

$2\|\hat{\boldsymbol{x}} - T\boldsymbol{x}^*\|_2 \cdot \sup_{\boldsymbol{v} \in (\mathcal{K}_\epsilon^-)^*} \frac{1}{m} \langle \boldsymbol{\eta}, \boldsymbol{Av} \rangle$, with the constraint set $(\mathcal{K}_\epsilon^-)^*$ defined in (3.2). Thus, in (3.3), in addition to $\mathscr{R}_u$, in the noisy setting we need to bound the additional term

$$\mathcal{R}_u' := \sup_{\boldsymbol{v} \in (\mathcal{K}_\epsilon^-)^*} \frac{1}{m} \langle \boldsymbol{\eta}, \boldsymbol{Av} \rangle = \sup_{\boldsymbol{v} \in (\mathcal{K}_\epsilon^-)^*} \frac{1}{m} \sum_{i=1}^m \eta_i \boldsymbol{a}_i^\top \boldsymbol{v}.$$

This can be done by the following lemma, which indicates that the sharp (uniform) rate in Theorem 1 can be retained in the presence of noise $\boldsymbol{\eta}$.

**Lemma 15.** (Bounding the additional term $\mathcal{R}_u'$). *In the noisy setting described above, with probability at least $1 - C_1 \exp(-\Omega(k \log \frac{TLr}{\epsilon})) - C_2 \exp(-\Omega(m))$, we have $\mathcal{R}_u' \lesssim \sqrt{\frac{k \log \frac{TLr}{\epsilon}}{m}}$.*

*Proof.* Conditioning on $\boldsymbol{\eta}$, the randomness of $\boldsymbol{a}_i$'s gives $\frac{1}{m} \sum_{i=1}^m \eta_i \boldsymbol{a}_i \sim \mathcal{N}(0, \frac{\|\boldsymbol{\eta}\|_2^2}{m^2} \boldsymbol{I}_n)$, and so $\|(\frac{1}{m} \sum_{i=1}^m \eta_i \boldsymbol{a}_i)^\top \boldsymbol{v}_1 - (\frac{1}{m} \sum_{i=1}^m \eta_i \boldsymbol{a}_i)^\top \boldsymbol{v}_2\|_{\psi_2} \le \frac{C_0 \|\boldsymbol{\eta}\|_2 \|\boldsymbol{v}_1 - \boldsymbol{v}_2\|_2}{m}$ holds for any $\boldsymbol{v}_1, \boldsymbol{v}_2 \in \mathbb{R}^n$. Let $\omega(\cdot)$ be the Gaussian width as defined in Lemma 5. Then, using the randomness of $\boldsymbol{a}_i$'s, Talagrand's comparison inequality [50, Exercise 8.6.5] yields that for any $t \ge 0$, we have

$$\mathbb{P}\left( \mathcal{R}_u' \le \frac{C_1 \|\boldsymbol{\eta}\|_2 \cdot [\omega((\mathcal{K}_\epsilon^-)^*) + t]}{m} \right) \ge 1 - 2 \exp(-t^2). \tag{F.1}$$

Next, we bound the Gaussian width $\omega((\mathcal{K}_\epsilon^-)^*)$. Recall that $(\mathcal{K}_\epsilon^-)^*$ is defined in (3.2), and Lemma 6 bounds its metric entropy as $\mathscr{H}((\mathcal{K}_\epsilon^-)^*, \eta) \le 2k \log \frac{12TLr}{\epsilon\eta}$. Thus, we can invoke Dudley's integral inequality [50, Theorem 8.1.3] to obtain

$$\omega((\mathcal{K}_\epsilon^-)^*) \le C_2 \int_0^2 \sqrt{2k \log \frac{12TLr}{\epsilon\eta}} \, \mathrm{d}\eta \lesssim \sqrt{k \log \frac{TLr}{\epsilon}}.$$

Now, we further let $t = \sqrt{k \log \frac{TLr}{\epsilon}}$ in (F.1) to obtain that $\mathcal{R}_u' \lesssim \frac{\|\boldsymbol{\eta}\|_2 \sqrt{k \log \frac{TLr}{\epsilon}}}{m}$ holds with probability at least $1 - 2 \exp(-k \log \frac{TLr}{\epsilon})$. It remains to deal with the randomness of $\boldsymbol{\eta}$ and bound $\|\boldsymbol{\eta}\|_2$. Because $\boldsymbol{\eta}$ has i.i.d. entries with $\|\eta_i\|_{\psi_2} = O(1)$, by [50, Theorem 3.1.1] we can obtain that $\|\boldsymbol{\eta}\|_2 \le C_3 \sqrt{m}$ with probability at least $1 - 2 \exp(-c_3 m)$. Substituting this bound into $\mathcal{R}_u' \lesssim \frac{\|\boldsymbol{\eta}\|_2 \sqrt{k \log \frac{TLr}{\epsilon}}}{m}$, the result follows. $\square$

To close this appendix, we briefly state how to adapt the proof of Theorem 1 to explicitly include the additive noise $\boldsymbol{\eta}$. Specifically, the left-hand side of (3.1) and its uniform lower bound $\Omega(\|\hat{\boldsymbol{x}} - T\boldsymbol{x}^*\|_2^2)$ remain unchanged, while the right-hand side of (3.1) is now bounded by $2\|\hat{\boldsymbol{x}} - T\boldsymbol{x}^*\|_2 \cdot (\mathscr{R}_{u1} + \mathscr{R}_{u2} + \mathcal{R}_u')$ (with $2\|\hat{\boldsymbol{x}} - T\boldsymbol{x}^*\|_2 \mathcal{R}_u'$ being the additional term); thus, combining the bound (C.12) on $2\|\hat{\boldsymbol{x}} - T\boldsymbol{x}^*\|_2 \cdot (\mathscr{R}_{u1} + \mathscr{R}_{u2})$ and Lemma 15, we establish a uniform upper bound

$$O\left( \|\hat{\boldsymbol{x}} - T\boldsymbol{x}^*\|_2 \cdot \left[ (A_g^{(1)} \vee A_g^{(2)}) \sqrt{\frac{k\mathscr{L}}{m}} + \sqrt{\frac{k \log \frac{TLr}{\epsilon}}{m}} \right] \right)$$

for the right-hand side of (3.1). Therefore, to ensure uniform recovery up to the $\ell_2$-norm accuracy of $\epsilon$ under the sub-Gaussian noise $\boldsymbol{\eta}$, it suffices to have a sample complexity

$$m \gtrsim (A_g^{(1)} \vee A_g^{(2)})^2 \frac{k\mathscr{L}}{\epsilon^2} + \frac{k \log \frac{TLr}{\epsilon}}{\epsilon^2}. \tag{F.2}$$

Since the logarithmic factors $\mathscr{L}$ in (C.11) dominates $\log \frac{TLr}{\epsilon}$, (F.2) indeed coincides with the sample complexity $m \gtrsim (A_g^{(1)} \vee A_g^{(2)})^2 \frac{k\mathscr{L}}{\epsilon^2}$ in Theorem 1 under the mild condition of $A_g^{(1)} \vee A_g^{(2)} = \Omega(1)$.

# G   Experimental Results for the MNIST dataset

## G.1   Details of the Settings

In this section, we conduct experiments on the MNIST dataset [28] to support our theoretical framework. We use various nonlinear measurement models, including 1-bit, dithered 1-bit, ReLU,

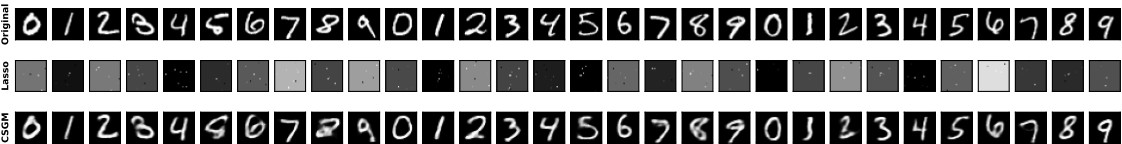

Figure 2: Reconstructed images of the MNIST dataset for the noiseless 1-bit measurements with $m = 150$.

and uniformly quantized CS with dithering (UQD). We select 30 images from the MNIST testing set, ensuring that there are three images from each of the 10 classes for maximum variability. A single measurement matrix $\boldsymbol{A}$ is generated and used for all 30 test images. All the experiments are repeated for 10 random trials. All the experiments are run using Python 3.10.6 and PyTorch 2.0.0, with an NVIDIA RTX 3060 Laptop 6GB GPU.

We train a variational autoencoder (VAE) on the training set of the MNIST dataset, which has 60,000 images, each of size 784. The decoder of the VAE is a fully connected neural network with ReLU activations, with input dimension $k = 20$ and output dimension $n = 784$, and two hidden layers with 500 neurons each. We train the VAE using the Adam optimizer with a mini-batch size of 100 and a learning rate of 0.001.

Since our contributions are primarily theoretical, we only provide simple proof-of-concept experimental results. In particular, since (2.1) is intractable to solve exactly, to estimate the underlying signal, we choose to use the algorithm proposed in [2] (referred to as CSGM) to approximate it. CSGM performs a gradient descent algorithm in the latent space in $\mathbb{R}^k$ with random restarts. In addition, we compare with the Lasso program that is solved by the iterative shrinkage thresholding algorithm.

For CSGM, we follow the setting in [2] and perform 10 random restarts with 1000 gradient descent steps per restart and pick the reconstruction with the best measurement error.

### G.2 Experimental Results for Noiseless 1-bit Measurements and Uniformly Quantized Measurements with Dithering

In this subsection, we present the numerical results for 1-bit measurements and uniformly quantized measurements with dithering, while the results for dithered 1-bit measurements and the Lipschitz SIM where the nonlinear link function is ReLU are similarly provided in Appendix G.3. For 1-bit measurements, since the underlying signal is assumed to be a unit vector and we aim to recover the direction of the signal, we use cosine similarity that is calculated as $\hat{\boldsymbol{x}}^T \boldsymbol{x}^* / (\|\hat{\boldsymbol{x}}\|_2 \cdot \|\boldsymbol{x}^*\|_2)$ with $\hat{\boldsymbol{x}}$ being the estimated vector to measure the reconstruction performance. For uniformly quantized measurements with dithering, we use the relative $\ell_2$-norm distance between the underlying signal and the estimated vector, i.e., $\|\hat{\boldsymbol{x}} - \boldsymbol{x}^*\| / \|\boldsymbol{x}^*\|_2$, to measure the reconstruction performance.

Since this paper is concerned with uniform recovery performance, in each trial, we record the worst-case reconstruction performance (i.e., the smallest cosine similarity or the largest relative error) over the 30 test images, and the worst-case cosine similarity or relative error is averaged over 10 trials.

Figures 2, 3, and 4 show that for noiseless 1-bit measurements and uniformly quantized measurements with dithering with $\delta = 3$, the CSGM approach can produce reasonably accurate reconstruction for all the test images when the number of measurements $m$ is as small as 150 and 100 respectively.

### G.3 Experimental Results for ReLU and Dithered 1-bit Measurements

We present the experimental results for the ReLU link function and dithered 1-bit measurements in Figures 5, 6, and 7. For dithered 1-bit measurements, we set $\lambda = R\sqrt{\log m}$ with $R > 0$ being a tuning parameter. For the case of using the ReLU link function, similarly to noiseless 1-bit measurements, we calculate the cosine similarity to measure the reconstruction performance. For dithered 1-bit measurements, similarly to uniformly quantized measurements with dithering, we calculate the relative $\ell_2$-norm distance. We observe that for these two nonlinear measurement models

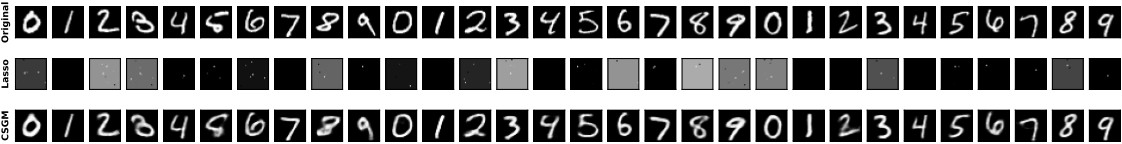

Figure 3: Reconstructed images of the MNIST dataset for UQD with $m = 100$ and $\delta = 3$.

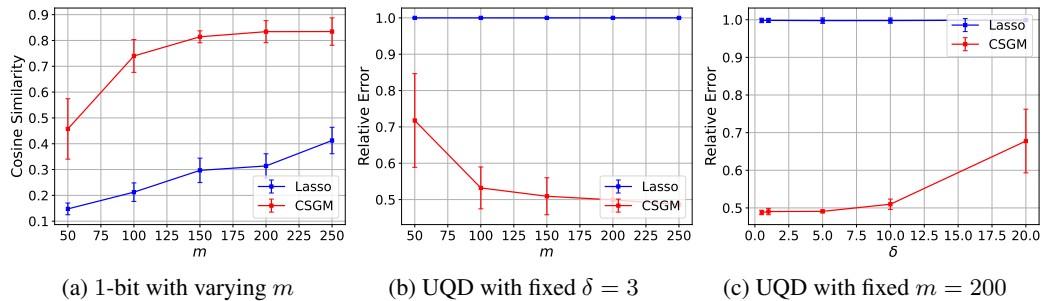

(a) 1-bit with varying $m$  (b) UQD with fixed $\delta = 3$  (c) UQD with fixed $m = 200$

Figure 4: Quantitative results of the performance of `CSGM` for 1-bit and UQD measurements on the MNIST dataset.

with a single realization of the random measurement ensemble, `CSGM` can also lead to reasonably good reconstruction for all the test images when the number of measurements is small compared to the ambient dimension.

## H Experimental Results for the CelebA dataset

In this section, we present numerical results for the CelebA dataset [35], which contains more than 200,000 face images for celebrities with an ambient dimension of $n = 12288$. We train a deep convolutional generative adversarial network (DCGAN) following the settings in `https://pytorch.org/tutorials/beginner/dcgan_faces_tutorial.html`. The latent dimension of the generator is $k = 100$ and the number of epochs for training is 20. Since the experiments for CelebA are more time-consuming than those of MNIST, we select 20 images from the test set of CelebA and perform 5 random trials. Other settings are the same as those for the MNIST dataset.

Since we have observed from the numerical results for MNIST that the experiments for the ReLU link function and dithered 1-bit measurements are similar, we only present the results for noiseless 1-bit measurements and uniformly quantized observations with dithering.

From Figures 8 and 10, we observe that for noiseless 1-bit measurements with 1500 samples, a single measurement matrix $\boldsymbol{A}$ can lead to reasonably accurate reconstruction for all the 20 test images. In addition, from Figures 9 and 10, we observe that for uniformly quantized measurements with

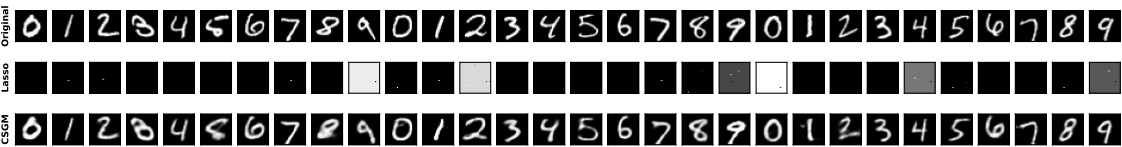

Figure 5: Reconstructed images of the MNIST dataset for the ReLU link function with $m = 150$ and $\sigma = 0.2$.

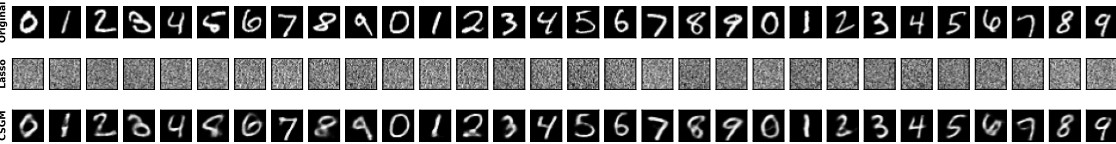

Figure 6: Examples of reconstructed images of the MNIST dataset for dithered 1-bit measurements with $m = 250$ and $R = 5$.

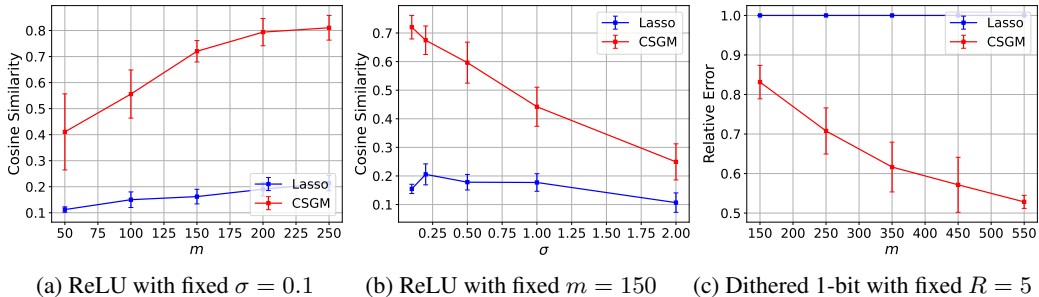

(a) ReLU with fixed $\sigma = 0.1$  (b) ReLU with fixed $m = 150$  (c) Dithered 1-bit with fixed $R = 5$

Figure 7: Quantitative results of the performance of CSGM for the ReLU link function and dithered 1-bit measurements on the MNIST dataset.

dithering, a single realization of the measurement matrix and random dither is sufficient for the reasonably accurate recovery of the 20 test images when $m = 1000$ and $\delta = 20$.

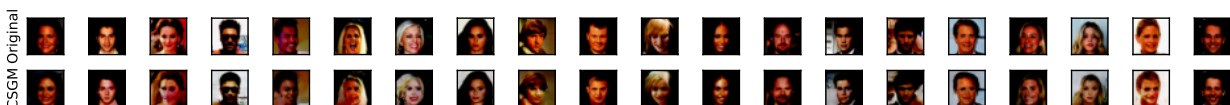

Figure 8: Reconstructed images of the CelebA dataset for the noiseless 1-bit measurements with $m = 1500$.

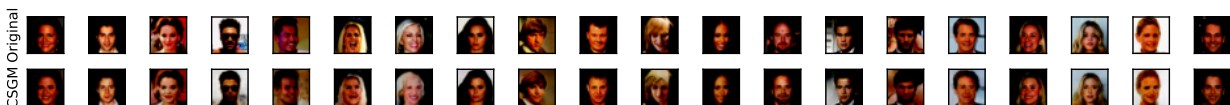

Figure 9: Reconstructed images of the CelebA dataset for UQD with $m = 1000$ and $\delta = 20$.

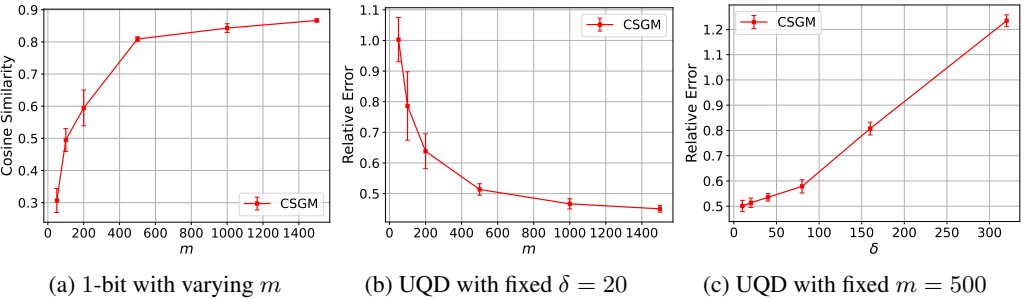

(a) 1-bit with varying $m$     (b) UQD with fixed $\delta = 20$     (c) UQD with fixed $m = 500$

Figure 10: Quantitative results of the performance of CSGM for 1-bit and UQD measurements on the CelebA dataset.

