# OpenReview forum: "A Unified Framework for Uniform Signal Recovery in Nonlinear Generative Compressed Sensing"
_NeurIPS.cc/2023/Conference — NeurIPS 2023 poster_

### Official Review · Reviewer_6eZp · 2023-07-05

**Soundness:** 3 good
**Presentation:** 2 fair
**Contribution:** 3 good
**Rating:** 6
**Confidence:** 4

**Summary:**

In non-linear compressed sensing, one would like to recover x from a
series of observations y_i = f_i(a_ix); and in generative compressed
sensing, x is drawn from a generative model.  The contribution of this
paper is a framework for uniform recovery with general nonlinear
measurements and Lipschitz generative models; in particular, this
framework includes Lipschitz generative models and dithered 1-bit
measurements, which were not previously known to give uniform
recovery.


**Strengths:**

The result is fairly general, handling dithered 1-bit or otherwise
discretized measurements, as well as noise in the measurements.  The
bound is pretty good, basically ideal except possibly for some terms
inside the log factor.


**Weaknesses:**

The first specific application the authors present for this framework
is:

 (1) uniform recovery of (2) Lipschitz generative models from (3)
 dithered 1-bit measurements

This is pretty specific, and if any one of the terms is relaxed
(nonuniform, a ReLU generator, or non-dithered) then prior work covers
it.  That is to say: the result is nice and general, but prior work
has pretty well covered the most interesting cases covered by this
result.

The writeup could be improved, repeatedly defining things in terms of
terms that are defined pages later.  For example, the main theorem
(Theorem 1) relies on script(L), which isn't defined until the
supplemental material; Assumption 3 uses section 3;


**Questions:**

* Can you give a corollary for handling Gaussian noise?

* What would you do for nonlinear sensing that's more nonlinear than discretization, e.g., sinusoidal or quadratic observations?

**Limitations:**

Yes.

---

> ### Author Rebuttal · Authors · 2023-08-08
>
> Thanks for your positive assessment of this paper and the useful comments and suggestions.  We respond regarding the writing quality in the general response to all reviews, and respond to the other points as follows.
>
> (**If any one of the terms is relaxed (nonuniform, a ReLU generator, or non-dithered) then prior work covers it**)
>
> Thanks for the comment. We agree that some prior results have been established if any one of the three terms is changed, but we also note that any of the three terms is non-trivial rather than a straightforward extension:
>
> - As explained in the paper, it is often significantly more challenging to establish uniform guarantees compared to non-uniform ones.
> - Compared to ReLU networks studied in Qiu et al. (2020), our Lipschitz generative model is more general and requires different proof techniques.
> - Dithered 1-bit measurements and non-dithered 1-bit measurements are of very different characteristics, e.g., the former allows for norm recovery while the latter cannot; this can be seen by comparing our Corollaries 1 and 2.
> - Besides the 1-bit cases, we also mention that our Corollary 4, regarding dithered uniformly quantized measurements, is of practical interest and novel. Specifically, in the literature, there is no prior (uniform/non-uniform) guarantee for GCS under such a quantizer. Also, in a situation where we are allowed to sample several bits from each measurement, the uniform quantizer could retain more information and hence may be preferable.
> - We highlight the significance of a unified framework, which allows us to clearly see the ingredients (i.e., our Assumptions 2-4) that lead to sharp uniform recovery guarantees. We believe that this is also positioned as an important contribution.
>
> S. Qiu et al. "Robust one-bit recovery via ReLU generative networks: Near-optimal statistical rate and global landscape analysis." In ICML, 2020.
>
> (**Can you give a corollary for handling Gaussian noise?**)
>
> Since the non-linearities can be possibly random, Gaussian noise can be encompassed in our results for single index models (see Corollary 3). For the remaining quantization models, we may consider the case of adding Gaussian noise explicitly with the measurement model $\mathbf{y}=f(\mathbf{A}\mathbf{x}^*)+\mathbf{e}$ where $\mathbf{e}$ follows the $m$-dimensional standard Gaussian distribution. For this case, we will need to bound an additional term $\sup_{\mathbf{v}} \sum_{i=1}^m e_i \mathbf{a}_i^\top \mathbf{v}$, which is not a product process and is easy to handle compared to the current $\mathscr{R}_u$ (defined in the equation before Line 301). We will add a corollary for handling Gaussian noise in the revised version.
>
>
> (**What would you do for nonlinear sensing that's more nonlinear than discretization, e.g., sinusoidal or quadratic observations?**)
>
> Since the $\sin(\cdot)$ function is 1-Lipschitz continuous, the sinusoidal model where $f_i(x) = \sin(x)$ is encompassed by our single index model result (Corollary 3) where we can achieve accurate recovery without knowing $f_i$. For the quadratic model where $f_i(x)=x^2$ (which corresponds to the phase retrieval problem), we note that the important parameter $T$  (see Line 138) is zero, which means our results are not applicable to this model. However, we note that it is a common issue that classical single index models do not encompass the quadratic model, and phase retrieval models are typically studied separately from our sort of models. See, e.g., Page 5 and the Conclusion Section of Yang et al. (2017). We thank the reviewer for this interesting question and we leave the uniform recovery guarantees for generative model based phase retrieval for future study.
>
> Z. Yang et al. "High-dimensional non-Gaussian single index models via thresholded score function estimation." In ICML, 2017.

---

> > ### Comment · Reviewer_6eZp · 2023-08-12
> >
> > Thanks for your response.  Would it be possible for you to give the corollary for Gaussian noise now?   I'm curious whether it is reasonably tight.

---

> > > ### Author Response · Authors · 2023-08-13
> > > **A Corollary for handling Gaussian noise and its proof**
> > >
> > > Thanks for your prompt response and the further question. Since it is a bit troublesome to display long equations in the OpenReview system, we provide the following anonymous link for the document regarding the corollary:
> > >
> > > **[EDIT: Link removed] Apologies, one author had missed the instruction not to include external links, so we have replicated the proof in a follow-up reply to this post instead**
> > >
> > > As we indicated in our initial response, we still have the sharp rate in the presence of Gaussian noise.
> > >
> > > The analysis is significantly simpler than that already done to handle other terms in our paper, so we view its addition as only a minor modification.  The level of detail for this proof will be expanded slightly for the revised paper.

---

> > > > ### Author Response · Authors · 2023-08-13
> > > > **Details of the Corollary**
> > > >
> > > > Apologies, one author had missed the instruction not to include external links, so we have replicated the proof below and will edit our previous response to remove the link.
> > > >
> > > > ===
> > > >
> > > > We consider the noisy model
> > > > $$\mathbf{y}=\mathbf{f}(\mathbf{Ax^*})+\mathbf{\epsilon},$$
> > > >
> > > >  where $\mathbf{\epsilon}\sim \mathcal{N}(0,\sigma^2\mathbf{I_m})$ is independent of $(\mathbf{A},\mathbf{f})$. Along similar lines as (3.1)-(3.3) in the paper, $\mathbf{\epsilon}$ leads to an additional term $\frac{2}{m}\big<\mathbf{\epsilon},\mathbf{A}(\mathbf{\hat{x}}-T\mathbf{x^*})\big>$ to the right-hand side of (3.1), and it suffices to bound
> > > > $$\\|\mathbf{\hat{x}}-T\mathbf{x^*}\\|_2 \cdot \sup\_{\mathbf{v}\in (\mathcal{K}\_{\epsilon}^-)^*} \frac{1}{m}\sum\_{i=1}^m \epsilon\_i \mathbf{a}\_i^T \mathbf{v},$$
> > > >
> > > > where the constraint set $(\mathcal{K}\_{\epsilon}^-)^*$ is defined in (3.2) of the paper.
> > > >
> > > > Thus, we can handle $\mathbf{\epsilon}$ by the following corollary while still retaining the sharp rate presented in the paper.  Here we let $\mathcal{R}_u'$ denote the above supremum.
> > > >
> > > > **Corollary:**
> > > >     $\mathcal{R}\_u' \lesssim \sigma\sqrt{\frac{k\log\frac{TLr}{\epsilon}}{m}}$ with   probability at least $1-C_1\exp(-\Omega(k\log\frac{TLr}{\epsilon}))-C_2\exp(-\Omega(m))$.
> > > >
> > > > **Proof:**
> > > >     Conditioning on $\mathbf{\epsilon}$, we have $\frac{1}{m}$ $\sum_{i=1}^m$ $\epsilon_i\mathbf{a_i}$ $\sim $ $\mathcal{N}$ $(0$,$\frac{\\|\mathbf{\epsilon}\\|_2^2}{m^2}$ $\mathbf{I_n})$. Thus, using the randomness of $\mathbf{a}_i$ and Talagrand's comparison inequality (e.g., Exercise 8.6.5 in Vershynin 2018), we know that for any $u>0$, with probability at least $1-2\exp(-u^2)$ we have
> > > > $$
> > > >         \mathcal{R}_u' \lesssim \frac{\\|\mathbf{\epsilon}\\|_2\cdot[\omega((\mathcal{K}\_{\epsilon}^-)^*)+u]}{m},
> > > > $$
> > > >
> > > > where $\omega(\mathcal{V})$ is the Gaussian width of $\mathcal{V}\subset \mathbb{R}^n$.
> > > >
> > > > Recall that $(\mathcal{K}_{\epsilon}^-)^*$ is defined in (3.2) of the paper, and we have shown $\mathcal{H}((\mathcal{K}\_{\epsilon}^-)^*,\eta)\leq 2k\log\frac{12TLr}{\epsilon\eta}$ in Lemma 6 of the paper, then by Dudley's integral inequality (see, e.g., Thm. 8.1.3 in Vershynin 2018) we have $\omega((\mathcal{K}\_{\epsilon}^-)^*)\lesssim\sqrt{k\log\frac{TLr}{\epsilon}}$.  Thus, we let $u=\sqrt{k\log\frac{TLr}{\epsilon}}$ in the above display to obtain that with probability at least $1-2\exp(-k\log\frac{TLr}{\epsilon})$, we have $\mathcal{R}\_u'\lesssim \frac{\\|\mathbf{\epsilon}\\|_2\sqrt{k\log\frac{TLr}{\epsilon}}}{m}$. Furthermore, for the randomness of $\mathbf{\epsilon}$, by  Thm. 3.1.1 in Vershynin 2018, $\\|\mathbf{\epsilon}\\|_2\lesssim \sigma\sqrt{m}$ holds with probability at least $1-\exp(-\Omega(m))$. Thus, the result follows.
> > > >
> > > > Please note that this analysis is significantly simpler than that already done to handle other terms on our paper, so we view its addition as only a minor modification.

---

### Official Review · Reviewer_85GG · 2023-07-05

**Soundness:** 2 fair
**Presentation:** 3 good
**Contribution:** 2 fair
**Rating:** 3
**Confidence:** 4

**Summary:**

This paper introduced a unified framework for uniform signal recovery in nonlinear generative compressed sensing, in particular, 1-bit generative compressed sensing (GCS) and single-index models (SIM).   The authors obtain  uniform recovery guarantees for 1-bit  GCS, 1-bit GCS with dithering, Lipschitz-continuous SIM, and uniformly quantized GCS with dithering.  Experimental results are presented to corroborate the theoretical results.

**Strengths:**

1.  A unified framework for uniform signal recovery in nonlinear generative compressed sensing is proposed.
2.  Uniform recovery guarantees are obtained for 1-bit  GCS, 1-bit GCS with dithering, Lipschitz-continuous SIM, and uniformly quantized GCS with dithering.



**Weaknesses:**

1. Both main statements and corresponding proofs build on the strong assumption that the target signal exactly belongs to the domain of generative models. Apparently, this is not the case for real-life compressed sensing, both theoretically and empirically. The key idea of generative compressed sensing is to recover an unknown target signal by leveraging the generative prior to capture its intrinsic structure. Nevertheless,  it does not assume that the target signal is exactly generated by the assumed generative models. However, the goal of this paper is to prove the uniform recovery of a signal that is exactly generated by a known generative model, which is apparently not the case of real compressed sensing, or is a different problem.  What if the target signal is not generated by the assumed generative model?


2. Given that this paper assumes that the target signal is exactly generated by a known generative model, how can the results be applied to practical generative models. For example, given two algorithms, one using a VAE prior, the other using a GAN prior, how to compare the corresponding reconstruction results?


3. Are the main results applicable to diffusion models? This paper assumes an L-Lipschitz continuous generative model and a radius r Ball in R^k. Recently, there are several recent studies on diffusion models for generalized linear inverse problems including 1-bit compressed sensing, for example:

[R1] Meng, Xiangming, and Yoshiyuki Kabashima. "Quantized Compressed Sensing with Score-Based Generative Models." ICLR2023
[R2] Chung, Hyungjin, Jeongsol Kim, Michael T. Mccann, Marc L. Klasky, and Jong Chul Ye. "Diffusion posterior sampling for general noisy inverse problems." ICLR2023.

[R1] discussed quantized compressed sensing with score-based generative models and [R2] introduces a unified diffusion posterior sampling method for general noisy linear and nonlinear problems. Can the main statements in this paper apply to the above latest studies with diffusion models? Please illustrate the main differences and explain why.




**Questions:**

See above

**Limitations:**

Unreasonable assumptions and lack of discussions of some latest works.

---

> ### Author Rebuttal · Authors · 2023-08-08
>
> Thanks for your useful comments and questions. Regarding the signal lying exactly in Range(G) and applying to practical generative models, please see the general response above.
>
> (**Are the main results applicable to diffusion models?**)
>
> In this paper, our focus is on generative models with a low-dimensional latent structure of size $k \ll n$, which captures many important generative models of interest.   However, diffusion models are very different in that they map $\mathbb{R}^n \to \mathbb{R}^n$ and create significant mathematical complications, e.g., Range(G) may be prohibitively large unless we only consider a small subset (e.g., one with high probability) of the latent space.  Being unable to handle such models when studying the sample complexity of high-dimensional inverse problems is not specifically a limitation of our work, but rather a **substantial open problem that is yet to be tackled by anyone** (to our knowledge).  Accordingly, we strongly believe that addressing this open problem should not be expected in a work of our nature.  To back up our claims, we note that even papers specifically about provable recovery with diffusion models do not give sample complexity guarantees (e.g., see https://arxiv.org/abs/2307.00619 and https://arxiv.org/abs/2302.01217).
>
> Nevertheless, we thank the reviewer for pointing out these two interesting papers on diffusion models. We will cite these papers in a revised Conclusion and Future Work Section and mention that providing similar uniform sample complexity guarantees for diffusion models is an interesting future direction.

---

> > ### Comment · Reviewer_85GG · 2023-08-16
> > **Response to rebuttal**
> >
> > Many thanks for the rebuttal.
> >
> > I do agree that such a theoretical analysis is interesting and might be non-trivial. My previous concern is that even a tiny relaxation of the strong assumption that the target signal exactly lies in Range(G) might lead to a fundamentally different result for uniform signal recovery. Moreover, the assumptions applied to the generative model in the proof might be difficult to verify for practical generative models. As a result, it is suggested to state the results carefully and state these limitations more clearly.
> >
> > In addition, from my understanding, the so-called "proof-of-concept experiments" does not "corroborate our theory" as stated.
> >
> > (1) The main results of the theory (Section 2.3) are about the relationship between the required sample complexity and a prescribed estimated error, while there is no such evaluation or verification in the experiments. I mean, I did not see how the presented results can support the main theoretical results of the paper.
> >
> > (2) In the theoretical analysis, the generalized Lasso is considered. However, in the experiment parts, CSGM is considered. Then, how can the experiments corroborate the theory? Why not use generalized Lasso in the experiments?
> >
> > Another concern is that the authors stated that the main results apply to the case when the observation model f is unknown. This result is a bit surprising since it is extremely challenging to recover x when f is unknown. How can it share the same result as the case when f is known?  I find it difficult to understand this from the current proof sketch in Section 3, nor did I see any experimental results for such a case in the appendix. Could the authors please explain this point a bit?

---

> > > ### Author Response · Authors · 2023-08-18
> > > **Follow-up response**
> > >
> > > Thanks for the response and further questions, which we clarify below.  We will also highlight the assumption of no representation error more clearly in the revised paper.
> > >
> > > **(Are the experiments running generalized Lasso?)**
> > >
> > > Please note that when we mentioned running CSGM, we meant using it to (approximately) solve Eq. (2.1), which is precisely the generalized Lasso.  Eq. (2.1) is intractable to solve exactly, and the CSGM approach is to approximate it using gradient descent with random restarts.  Previously CSGM was devised specifically for linear models, but generalized Lasso contains the exact same objective function and constraint (up to scaling by $T$; see Remark 5) as the original one for linear models, so CSGM remains applicable despite the non-linearity.
> > >
> > > This idea of approximating the exact optimization by gradient methods is extremely standard in the literature on CS with generative priors, e.g., see Bora et al. (2017), Dhar et al. (2018), and Liu et al. (2020) in our paper’s reference list.  Thus, up to very standard practical approximations, **we are running generalized Lasso**.  We will re-word to make this clearer.
> > >
> > > **(Do the experiments corroborate the theory?)**
> > >
> > > We believe that the previous response partially addresses this, but provide further discussion as follows.
> > >
> > > We chose the word “corroborate” to avoid overly strong language like “verify” or “confirm”, but we would be happy to tone this down further and replace it by a more precise statement, e.g., “to demonstrate that a practical variant of the generalized Lasso can be effective in recovering multiple signals with a single measurement matrix.”.
> > >
> > > Confirming the scaling laws experimentally is a challenging task, and to our knowledge it has never been attempted in the literature on CS with generative priors.  Our experiments are aligned with those performed in similar kinds of works that consider generative priors (e.g., those mentioned above), with the distinction that we take the *worst case* performance over *batches* of images to better align with our goal of uniform recovery.  With the above-mentioned re-wording and the removal of any suggestion that we are verifying our theory, we believe that they are a useful addition to the paper.
> > >
> > > Having said this, we emphasize that **by far** our main contributions are our theoretical results, so we hope that they will accordingly be the main factor in the final decision.  (For comparison, other works on single index models (SIMs) typically have no experiments at all, e.g., Plan & Vershynin (2016), Genzel (2016), and the most relevant work Genzel & Stollenwerk (2023).)
> > >
> > > **(How can unknown $f$ be possible?)**
> > >
> > > From an experimental point of view, the fact that we run CSGM to approximate Eq. (2.1) and get good results supports the fact that this is possible.
> > >
> > > From a mathematical point of view, existing works demonstrated that the SIM with unknown nonlinearity $f$ (as considered in Part C of our Section 2.3) can be “transformed” into a linear model with an “unconventional noise term”.  Specifically, Plan & Vershynin (2016)'s Section 4 demonstrates that if $f$ satisfies the condition that $\mu := \mathbb{E}_{g \sim \mathcal{N}(0,1)}[f(g)g] \ne 0$, then $\mathbf{y} = f(\mathbf{A}\mathbf{x}^*)$ (where $f$ is applied element-wise and $\mathbf{A}$ has standard Gaussian entries) can be written as $\mathbf{y}=\mathbf{A}\mu\mathbf{x}^*+\mathbf{w}$, with $\mathbf{w}$ satisfying $\mathbb{E}[\mathbf{A}^\top\mathbf{w}]=\mathbf{0}$ thus acting as an unconventional noise vector.  Although the generalized Lasso approach is most naturally suited to conventional noise such as Gaussian, it turns out to still work under this unconventional noise.
> > >
> > >
> > > We omitted the above discussion because it is already well-documented in previous works, but we would be happy to use the extra available page (if accepted) to include an overview similar to the above paragraph.

---

> > > > ### Comment · Reviewer_85GG · 2023-08-19
> > > > **Response to follow-up response**
> > > >
> > > > I appreciate the authors' response.
> > > >
> > > > 1. I got your meaning regarding the relation between generalized Lasso and CSGM. Thanks for the explanation. I highly suggest clarifying this point since its current description is a bit confusing.
> > > >
> > > > 2. Regarding whether the experiments corroborate the theory, I still strongly believe that the provided experiments are far from the theoretical statement. The lack of such experiments in the literature on CS with generative priors can not be an excuse for not doing so since the main focuses are different, i..e, previous work does not provide the same claim as this paper. The authors have to provide demonstrations of their own contributions.  From another perspective, this concern is kind of related to what I meant in the original comments of weakness 2 on the practicality of the result. In addition, the experiments for scaling law analysis are not that difficult to design, as typically shown in Ravikumar, Pradeep, Martin J. Wainwright, and John D. Lafferty. "High-dimensional Ising model selection using l1-regularized logistic regression." (2010): The Annals of Statistics, 1287-1319.
> > > >
> > > > 3. Regarding the case of unknown f, the success of Lasso in the nonlinear case is a well-known result. Nevertheless, I think it is essential to incorporate detailed rigorous proofs if the authors want to claim the result for unknown f in their setting with generative prior. The empirical success of CSGM for a few specific f cannot support such a general result of unknown f. Explicit conditions  (including corresponding proofs) on the unknown f should be given in the main theoretical results similar to Plan & Vershynin (2016). Moreover, I do not agree with the argument "We omitted the above discussion because it is already well-documented in previous works", since previous works do not consider the case with generative prior. Otherwise, in the authors' logic, the whole study of this paper can also be omitted since results with conventional prior (rather than generative prior) have already been studied and known in previous works.
> > > >
> > > > Overall, I agree that this paper is interesting and the authors have made inspiring progress,  but it is not ready enough for publication in NeurIPS in its current form due to the above concerns (both theoretical concerns and experimental concerns).

---

> > > > > ### Author Response · Authors · 2023-08-20
> > > > >
> > > > > Thank you for the additional feedback; we will keep our responses brief.
> > > > >
> > > > > *“I got your meaning regarding the relation between generalized Lasso and CSGM. Thanks for the explanation. I highly suggest clarifying this point since its current description is a bit confusing.”*
> > > > > - **We agree that this point will benefit from further clarification, and we will do so in the revised paper.**
> > > > >
> > > > > *“Regarding whether the experiments corroborate the theory, I still strongly believe that the provided experiments are far from the theoretical statement. The lack of such experiments in the literature on CS with generative priors can not be an excuse for not doing so since the main focuses are different, i..e, previous work does not provide the same claim as yours.”*
> > > > > - **We reiterate that we have agreed to avoid the phrase “corroborate our theory”.  Regarding the previous works, these works also provide sample complexity bounds – albeit under more specific observation models and/or for non-uniform recovery  – so we believe that comparing to them in our response was reasonable.  They of course don’t provide exactly the same claims, but they are still works of a very similar nature and focus.**
> > > > >
> > > > > *“The authors have to provide demonstrations of their unique contributions.  From another perspective, this is kind of related to what I meant in the original comments of weakness 2 on the practicality of the result. What can we truly expect to obtain from the main results since we already know that CSGM can well reconstruct the signals, without the theoretical results provided here?”*
> > > > > - **Our unique contribution is our theoretical framework, and we disagree with the idea that all theory must be verified experimentally (many important theory works do not do so).  Uniformity and non-linearity are highly sought-after notions in CS, e.g., see Qiu et al. [43].**
> > > > >
> > > > > *“The experiments for scaling law analysis are not that difficult to design, as typically shown in Ravikumar, Pradeep, Martin J. Wainwright, and John D. Lafferty. "High-dimensional Ising model selection using l1-regularized logistic regression." (2010): The Annals of Statistics, 1287-1319.”*
> > > > > - **Thank you for the reference, though we still believe there is major significance in the extensive recent work on generative priors, none of which have done this (to our knowledge).**
> > > > >
> > > > > *”Regarding the case of unknown f, the success of Lasso in the nonlinear case is a well-known result. Nevertheless, I think it is essential to incorporate detailed rigorous proof of this part if the authors want to claim the result for unknown f in their setting with generative prior.”*
> > > > > - **We are unsure what the reviewer means by this comment, as all of our theoretical results are all rigorously proved – the main Theorem in Appendix C, the corollaries in Appendix E, and auxiliary lemmas in Appendix D.**
> > > > >
> > > > > *“The success of CSGM for a few specific f cannot support such a general result of unknown f. Explicit conditions (including corresponding proof) on the unknown f should be given in the main theoretical results similar to Plan & Vershynin (2016).”*
> > > > > - **Again we are unsure what the reviewer means, as all the conditions on $f$ are stated in our main results via cross-references to Assumptions 1-4.  In Appendix E, we verify these assumptions for all the special cases that we apply our main theorem to.**
> > > > >
> > > > > *“Moreover, I do not agree with the argument "We omitted the above discussion because it is already well-documented in previous works", since previous works do not consider the case with generative prior. Otherwise, in the authors' logic, the whole study of this paper can also be omitted since results with conventional prior (rather than generative prior) have already been studied and known in previous works.”*
> > > > > - **We do not agree that our logic leads to such a statement.  The motivation/intuition of the estimator itself is similar regardless of the specific prior used, which is why we found it less important to repeat (though as we said, we will re-iterate it in the revised paper).  The same reasoning cannot be applied to the theorems/corollaries/proofs because they are far from being a direct consequence of existing ones.**

---

### Official Review · Reviewer_K1vm · 2023-07-06

**Soundness:** 3 good
**Presentation:** 2 fair
**Contribution:** 3 good
**Rating:** 6
**Confidence:** 3

**Summary:**

The paper discusses a unified framework for uniform signal recovery in nonlinear generative compressed sensing. The authors utilized the use of generalized Lasso and Lipschitz approximation to allow for a lower sample size of measurements.

**Strengths:**

In what follows the strengths of the paper are given:

1) The paper presents a framework for deriving uniform recovery guarantees for nonlinear generative compressed sensing where the observation model is nonlinear and possibly discontinuous or unknown.
2) Utilizing the generalized Lasso and Lipschitz approximation allowed for a lower sample size of $\tilde{O} \left( \frac{k}{\varepsilon^2} \right)$, which is smaller than the previously best bound of $\tilde{O}\left( \frac{k}{\varepsilon^4}\right)$.
3) The idea of drawing back from using a concentration inequality (Lemma 5) in the appendix and instead leveraging the use of metric entropy to derive a tighter upper bound on $\mathcal{R}_u$ is quite interesting.
4) Finally, the paper is mainly theoretical and the experimentation done in the supplementary is to show confirm the findings of the paper, which are extensively shown.

**Weaknesses:**

The paper is somewhat hard to follow:
1) Some notations are used before they are defined.
2) Some explanations around the mathematical parts are not properly stated, e.g., transitions.


**Questions:**

1) Please expand on the transitions between equations.
2) How did you ensure that the assumptions that were presented in the paper hold in your experiments?

**Limitations:**

The limitations are adequately addressed.

---

> ### Author Rebuttal · Authors · 2023-08-08
>
> Thanks for your recognition of this paper and the helpful comments. Regarding readability, please see the general response above.
>
> (**How did you ensure that the assumptions that were presented in the paper hold in your experiments?**)
>
> We would like to highlight that this work is primarily theoretical, and the experiments are basic proof-of-concept and not a main contribution. Assumptions 2-4 are assumed solely for the theory, and we have verified in Corollaries 1-4 (with the details being provided in Appendix E) that these assumptions are reasonable and are satisfied by various nonlinear models. As for Assumption 1, in particular for the assumption of no representation error (i.e., the target signal is contained in the range of the generative model) and its practical effect, please refer to our general responses to all reviewers.

---

### Author Rebuttal · Authors · 2023-08-08

**General responses to the three anonymous reviewers**

We are very grateful to the reviewers for their helpful feedback and suggestions. Our responses to the main concerns shared by multiple reviewers are given as follows. Other responses are given to each reviewer separately.

(**The assumption that the target signal exactly belongs to the range of generative models**)

We agree that relaxing the assumption of no representation error (i.e., the target signal lies exactly in the range of the generative model) is of significant interest. However, for high-dimensional single index models (SIMs) with the generalized Lasso method, it may be infeasible to obtain comparable theoretical guarantees upon doing so. To the best of our knowledge, all prior works in this line (high-dimensional SIMs with generalized Lasso type approaches) assume no representation error, even under simpler classical priors where the target signal is assumed to be exactly contained in a low-complexity structured set or for weaker non-uniform recovery guarantees. A partial reference list is provided below.

In particular, appropriately handling the representation error for SIMs (with the generalized Lasso approach) has been mentioned as an open problem in the Discussion Section of the seminal work by Plan & Vershynin (2016).  We also highlight the recent work of Genzel & Stollenwerk (2023), which is the most relevant work to ours and also considers a setting where the signal lies *exactly* in a known structured set.

As is evident from the fact that these recent papers have been published in the topmost venues in machine learning, and/or as evidenced by the substantial citation counts of these works (also shown on the list), we believe that the assumption of no representation error has been widely accepted in the active research area of high-dimensional SIMs.  We sincerely hope that the final score/decision for our submission will be based on the main goal and contributions of this particular paper, rather than on the general limitations in a broad and popular line of works.

References:

Y. Plan &  R. Vershynin. "The generalized lasso with non-linear observations." IEEE Trans. Inf. Theory, 2016. [198 citations]

M. Genzel. "High-dimensional estimation of structured signals from non-linear observations with general convex loss functions." IEEE Trans. Inf. Theory, 2016. [47 citations]

Z. Yang et al. "High-dimensional non-Gaussian single index models via thresholded score function estimation." In ICML, 2017. [47 citations]

X. Wei et al. "On the statistical rate of nonlinear recovery in generative models with heavy-tailed data." In ICML, 2019. [22 citations]

C. Thrampoulidis & A.S. Rawat. "The generalized lasso for sub-gaussian measurements with dithered quantization." IEEE Trans. Inf. Theory, 2020. [24 citations]

Z. Liu &  J. Scarlett. "The generalized lasso with nonlinear observations and generative priors." In NeurIPS, 2020. [12 citations]

M. Genzel &  A. Stollenwerk. "A unified approach to uniform signal recovery from nonlinear observations." Foundations of Computational Mathematics, 2023. [New paper]

(**Given the assumption that the target signal is exactly generated by a known generative model, how can the results be applied to practical generative models?**)

We would like to highlight that this work is primarily theoretical, and the experiments are basic proof-of-concept and not a main contribution. We observe from the experimental results presented in the supplementary material that we can obtain accurate reconstruction for MNIST and CelebA images using a relatively small number of samples. This indicates that the generalized Lasso approach itself is still effective with representation error, although the theory is an interesting open problem.

(**The paper is somewhat hard to follow or the writeup could be improved**)

In the revised paper, we will ensure that all the notations are defined before they are used and add more explanations between equations to make the paper easier to follow. This will include the specific suggestions by Reviewer 6eZp, and those we find from our own careful proofreading. We also welcome any further specific pointers from all three reviewers. While we will strive to be meticulous with these edits, we are confident that they will still amount to relatively minor changes.

---

### Decision · Program_Chairs · 2023-09-21

**Decision:**

Accept (poster)

**Comment:**

The paper studies non-linear compressed sensing, where one tries to recover a signal from non-linear measurements. This paper proposes a unified framework when the generative model is Lipschitz, which is a natural assumption. There are cases where previous results were not covering, e.g. Lipschitz generators and dithered 1-bit measurements, a very challenging case. Overall the paper is technically very strong and makes progress in the theory of compressed sensing with generative priors with complicated measurement models.

The authors did a very good job in extensively answering questions of the reviewers, including providing corollaries that were not clearly stated in the paper.

One reviewer raised several concerns about the limitation of the assumption that the signals indeed come from the generative model, but this is a standard assumption when developing theory in this space. Realizability is sensible to study if nothing has been known before for this type of measurement. Further the experimental evaluation is very limited in this paper as pointed out by reviewers. I still recommend that this paper is accepted since it makes serious project on a challenging and general inverse problem and the theoretical contributions are solid.